# Efficient Multivariate Robust Mean Estimation Under Mean-Shift Contamination

**Ilias Diakonikolas** [1]   **Giannis Iakovidis** [1]   **Daniel Kane** [2]   **Thanasis Pittas** [1]

## Abstract

We study the algorithmic problem of robust mean estimation of an identity covariance Gaussian in the presence of mean-shift contamination. In this contamination model, we are given a set of points in $\mathbb{R}^d$ generated i.i.d. via the following process. For a parameter $\alpha < 1/2$, the $i$-th sample $x_i$ is obtained as follows: with probability $1 - \alpha$, $x_i$ is drawn from $\mathcal{N}(\mu, I)$, where $\mu \in \mathbb{R}^d$ is the target mean; and with probability $\alpha$, $x_i$ is drawn from $\mathcal{N}(z_i, I)$, where $z_i$ is unknown and potentially arbitrary. Prior work characterized the information-theoretic limits of this task. Specifically, it was shown that— in contrast to Huber contamination— in the presence of mean-shift contamination consistent estimation is possible. On the other hand, all known robust estimators in the mean-shift model have running times exponential in the dimension. Here we give the first computationally efficient algorithm for high-dimensional robust mean estimation with mean-shift contamination that can tolerate a constant fraction of outliers. In particular, our algorithm has near-optimal sample complexity, runs in sample-polynomial time, and approximates the target mean to *any* desired accuracy. Conceptually, our result contributes to a growing body of work that studies inference with respect to natural noise models lying in between fully adversarial and random settings.

## 1. Introduction

Robust statistics (Huber & Ronchetti, 2009; Diakonikolas & Kane, 2023) aims to develop accurate estimators in the presence of a constant fraction of outliers. In a range of machine learning scenarios, the standard i.i.d. assumption does not accurately represent the underlying phenomenon. For example, in ML security applications (Barreno et al., 2010; Biggio et al., 2012; Steinhardt et al., 2017; Tran et al., 2018; Diakonikolas et al., 2019), the data may be adversarially manipulated or contain out-of-distribution data which arise from unknown categories (Du et al., 2024); in biological applications, datasets often contain natural outliers (Rosenberg et al., 2002; Paschou et al., 2010; Li et al., 2008) that may pollute downstream statistical analysis. The field of robust statistics originates from the early 1960s with the pioneering works of Tukey and Huber (Tukey, 1960; Huber, 1964). Early work in the field obtained minimax optimal robust estimators for the mean estimation and other tasks. However, the multivariate versions of these robust estimators incurred exponential runtime in the dimension. A recent line of work in computer science, starting with Diakonikolas et al. (2016); Lai et al. (2016), has led to a revival of robust statistics from an algorithmic standpoint, by providing the first robust estimators in high dimensions with polynomial sample and time complexity.

The prototypical robust statistics task is that of Gaussian mean estimation in Huber's contamination model (Huber, 1964). Letting $\alpha \in (0, 1/2)$ be the contamination parameter, each sample in Huber's model is drawn either from an inlier normal distribution $\mathcal{N}(\mu, I)$ (where $\mu$ is the unknown target mean) with probability $1 - \alpha$; or from an unknown and potentially arbitrary distribution $E$ with probability $\alpha$. Recent work has provided efficient algorithms with near-optimal guarantees in Huber's model (Diakonikolas et al., 2018a; 2024). Huber's contamination is a rather strong model, as it allows for the unknown distribution of corruptions to be arbitrary. On the one hand, this level of generality makes the model quite powerful—by allowing it to cover a wider range of phenomena. On the other hand, its definition is inherently tied to significant information-theoretic limitations. Specifically, it is known that even with an infinite number of samples, any mean estimator $\widehat{\mu}$ in Huber's model has to incur an error of $\|\mu - \widehat{\mu}\| = \Omega(\alpha)$, where $\alpha$ is the rate of contamination. This happens because it is possible to start from two Gaussians (even in one dimension) with means $\Omega(\alpha)$ apart, and mix them with appropriate outlier distributions so that the contaminated distributions are identical; see, e.g., Chapter 1 of Diakonikolas & Kane (2023).

Authors are listed in alphabetical order.  [1]University of Wisconsin-Madison [2]University of California, San Diego. Correspondence to: Giannis Iakovidis <iakovidis@wisc.edu>, Thanasis Pittas <pittas@wisc.edu>.

*Proceedings of the 42$^{st}$ International Conference on Machine Learning*, Vancouver, Canada. PMLR 267, 2025. Copyright 2025 by the author(s).

(Consistency is similarly unachievable in other commonly used corruption models like total variation contamination.)

A natural way to achieve consistency is to impose additional structure on the contamination model. A prominent and well-studied assumption is that outliers do not follow arbitrary distributions, but instead correspond to mean-shifted copies of the original distribution. Mean shift contamination has been extensively studied in the robust statistics literature, for both regression (Sardy et al., 2001; Gannaz, 2007; McCann & Welsch, 2007; She & Owen, 2011) and mean estimation (Cai & Jin, 2010; Collier & Dalalyan, 2019; Carpentier et al., 2021; Li, 2023; Kotekal & Gao, 2025).

In this paper, we consider arguably the most basic version of mean shift contamination, where the inliers are drawn from a known covariance Gaussian distribution with unknown mean $\mu$, while each outlier is sampled from a (potentially different) Gaussian with the same covariance and unknown arbitrary mean. The goal is to understand the statistical-computational landscape of mean estimation in this model.

**Definition 1.1** (Mean-Shift Contamination Model). *For $\alpha \in (0, 1/2)$ and $n \in \mathbb{Z}_+$, a set $T$ of $n$ points in $\mathbb{R}^d$ is called an $\alpha$-corrupted set of points from $\mathcal{N}(\mu, I)$ under the mean-shift model, if an adversary chooses $z_1, \ldots, z_n$, and each point $x_i \in T$ is then sampled independently as follows: With probability $1 - \alpha$, $x_i$ is sampled from $\mathcal{N}(\mu, I)$, and probability $\alpha$, $x_i$ is sampled from $\mathcal{N}(z_i, I)$.*

Some comments are in order. A more general version of Definition 1.1 would allow the inliers and outliers be drawn from Gaussians with covariance matrix $\Sigma$. When $\Sigma$ is known to the algorithm, it suffices to consider the case $\Sigma = I$ (as we can transform the samples and reduce to this case). The known covariance version of the shift contamination model has been studied in a number of works, including Collier & Dalalyan (2019); Carpentier et al. (2021); Li (2023); Kotekal & Gao (2025).[1] Moreover, the closely related (more challenging) model where the covariance is unknown has also been considered (Cai & Jin, 2010; Carpentier et al., 2021; Kotekal & Gao, 2025). Mean shift contamination has been studied from a theoretical statistics standpoint, with a focus on minimax rates, and as a modeling assumption (Jin & Cai, 2007; Sun & Cai, 2007; Cai & Sun, 2009; Efron, 2004; 2007; 2008). Additional motivation can be found in (Kotekal & Gao, 2025). A more detailed summary of related work is provided in Appendix A.

The majority of prior work has primarily focused on the one-dimensional setting. Specifically, Kotekal & Gao (2025) obtained sharp minimax rates (upper and lower bounds) of estimating $\mu$ in absolute error. (These works also consider the more general estimation task for the unknown vari-

---

[1] Carpentier et al. (2021) considers a special case of the model where $z_i - \mu$ are assumed to be non-negative.

ance case.) Intriguingly, and in sharp contrast to Huber's model, consistency is achievable in the mean-shift model. Intuitively, this happens because now all the samples are convoluted with a Gaussian; thus, given infinitely many samples, one can form the underlying distribution, perform deconvolution, and recover the single spike corresponding to the inliers. More specifically, Kotekal & Gao (2025) showed that, if the contamination parameter $\alpha$ is a positive constant strictly less than $1/2$, the mean can be estimated to any desired accuracy $\epsilon$ using $2^{\Theta(1/\epsilon^2)}$ samples. This can be used to show (see Proposition 2.1) that the $d$-dimensional version of the problem can be solved, up to $\ell_2$-error $\epsilon$, with $n = d\, 2^{O(1/\epsilon^2)}$ samples—an upper bound that is essentially best possible.

While the one-dimensional estimators of Li (2023); Kotekal & Gao (2025) can be implemented in polynomial time, very little is known from an algorithmic standpoint for the multivariate problem. Specifically, the only known methods (achieving any desired accuracy) involve a brute-force search with complexity that scales exponentially in the dimension (Proposition 2.1). Of course, one could apply efficient robust algorithms designed for Huber's model from the existing robust statistics literature. Unfortunately, such algorithms inherently cannot obtain error better than $\Omega(\alpha)$—while our goal is to achieve any desired accuracy $\epsilon \ll \alpha$. This discussion leads to the following question, which was the main motivation for this work:

*Is there a sample and computationally efficient multivariate robust mean estimator in the mean-shift model?*

By the term "sample efficient", we mean an estimator with sample complexity $n = \mathrm{poly}(d, 2^{1/\epsilon^2})$ samples. As is standard, computationally efficient refers to an algorithm with $\mathrm{poly}(n, d)$ runtime. In this paper, we answer this question in the affirmative. As a bonus, the sample complexity of our algorithm is minimax near-optimal, within logarithmic factors.

Specifically, we establish the following theorem:

**Theorem 1.2.** (Main Algorithmic Result) *Let $d \in \mathbb{Z}_+$ denote the dimension, $\mu \in \mathbb{R}^d$ be an unknown mean vector, $\epsilon \in (0, 1)$ be an accuracy parameter, and $\alpha \leq 0.49$ be a contamination parameter. There exists an algorithm that takes as input $\epsilon$, draws $n = \tilde{O}(d/\epsilon^{2+o(1)} + 2^{O(1/\epsilon^2)})$ $\alpha$-corrupted samples from $\mathcal{N}(\mu, I)$ under the mean-shift model (Definition 1.1), runs in $\mathrm{poly}(n, d)$ time, and outputs $\widehat{\mu}$ such that with probability at least $0.99$ it holds $\|\widehat{\mu} - \mu\| \leq \epsilon$.*

Theorem 1.2 gives the first sample efficient algorithm for our problem that runs in sample-polynomial time. We remind the reader that, in the Huber contamination model, this goal is information-theoretically impossible; that is, no algorithm can achieve consistency regardless of its sample and computational resources. Note that the sample com-

plexity of our algorithm is close to optimal: even for one-dimension with $\alpha \leq 0.49$, any estimator requires $2^{\Omega(1/\epsilon^2)}$ samples (Carpentier et al., 2021; Kotekal & Gao, 2025); and linear dependence on $d$ is necessary even for the outlier-free setting. Moreover, we point out that our algorithm does not need to know the value of the contamination parameter $\alpha$. While we have assumed for simplicity of the statement that $\alpha \leq 0.49$, we show in Appendix F (see Theorem F.1) that by a mild increase in the error and sample complexity, our algorithm can be generalized to work for $\alpha \in (0, 1/2 - c)$, for any constant $c \in (0, 1/2)$, without the need to know $c$ a priori.

## 1.1. Organization

The paper is structured as follows: In Section 1.2, we provide intuition about the techniques used and the structure of the algorithm. In Section 2, we formally state the algorithm and sketch its analysis. Finally, in Section 3, we present our conclusions and discuss some interesting open problems.

## 1.2. Our Techniques

Given samples of the form $x \sim \mathcal{N}(m, I)$, where $m$ is a random variable taking the value $m = \mu$ with probability $2/3$ (instead of $2/3$, our model of Definition 1.1 uses $1 - \alpha$; this is a simplification for the purpose of this proof overview), the goal is to estimate $\mu$ to $\ell_2$-error $\epsilon$ with high constant probability.

Note that, using the results of Li (2023); Kotekal & Gao (2025), it is straightforward to design a (computationally inefficient) algorithm with sample complexity $d2^{O(1/\epsilon^2)}$ and runtime $2^{O(d+1/\epsilon^2)}$ — via a standard reduction to the 1-dimensional case (cf. Proposition 2.1). In particular, for every $v$ in an exponentially-sized cover of the unit sphere, we apply a robust one-dimensional estimator to the projections of the samples $v^\top x$. This gives us an $\epsilon$-approximation of $v^\top \mu$. Piecing these values together for various values of $v$ allows us to reconstruct an $\epsilon$-approximation of $\mu$.

As a first step towards Theorem 1.2, we will design an algorithm with sample complexity $\text{poly}(d)2^{\Theta(1/\epsilon^2)}$ and runtime $\text{poly}(d)2^{\Theta(1/\epsilon^4)}$. Note that this is not yet a *computationally efficient* algorithm, since the runtime is *quasi-polynomial* in the sample size. It turns out that this simpler algorithm is a good start, as it will lay down the main ideas which eventually lead to the efficient algorithm of Theorem 1.2. The main idea is to design some kind of (data dependent) dimension reduction. Ideally, we would like to first find the direction $\mu/\|\mu\|$ along which $\mu$ lies, and then run a one-dimensional robust estimator on that direction to recover $\|\mu\|$. More realistically, we will attempt to recover a low-dimensional subspace that on which $\mu$ has a large projection, and then run a low-dimensional robust

estimator on that subspace. In particular, if we can find a subspace $\mathcal{V}$ of dimension $\dim(\mathcal{V}) \leq \text{poly}(1/\epsilon)$ such that $\|\text{Proj}_{\mathcal{V}^\perp}(\mu)\| \leq \epsilon/2$, then we can apply the algorithm of Proposition 2.1 to the projection of the dataset onto $\mathcal{V}$ to find a $\widehat{\mu} \in \mathcal{V}$ with $\|\text{Proj}_{\mathcal{V}}(\mu) - \widehat{\mu}\| \leq \epsilon/2$. This would imply the desired error bound of $\|\mu - \widehat{\mu}\| \leq \epsilon$.

A standard way to find such a $\mathcal{V}$ is by leveraging the second moment of the data. We have that $\mathbf{E}[xx^\top - I] = \mathbf{E}[mm^\top] \succcurlyeq \frac{2}{3}\mu\mu^\top$, since at least $2/3$ of samples (inliers) follow $\mathcal{N}(\mu, I)$ and the terms corresponding to outliers are positive semidefinite. Therefore, if $\mathcal{V}$ is the subspace spanned by all of the eigenvectors of $\mathbf{E}[xx^\top - I]$ with eigenvalues more than $\epsilon^2/6$, then $\|\text{Proj}_{\mathcal{V}^\perp}(\mu)\| \leq \epsilon/2$. Unfortunately, as $m$ is unbounded, $\mathbf{E}[mm^\top]$ could have many such eigenvalues. To fix this issue, we will need a way to effectively truncate the larger values of $m$.

To achieve this, we instead estimate the matrix $A := \mathbf{E}[(xx^\top - \frac{1}{1+2\gamma}I) \exp(-(\|x\|^2 - d)\gamma)]$, for some carefully chosen $\gamma < 1$. We can explicitly calculate the expectation for $x \sim \mathcal{N}(m, I)$, which turns out to be roughly equal to $mm^\top e^{-\|m\|^2\gamma}$ (cf. Lemma 2.3). Since $2/3$ fraction of the samples have $m = \mu$, it follows that $A \succcurlyeq \frac{2}{3}\mu\mu^\top e^{-\|\mu\|^2\gamma}$. Note that so long as $\|\mu\|$ is not too large (indeed, we can assume it is $O(1)$ by some naïve outlier removal), every unit vector with $v^\top Av \leq \epsilon^2/6$ must satisfy $|v^\top \mu| \leq \epsilon/2$ (cf. Lemma 2.9). Additionally, it is not hard to see that $A$ has trace bounded by $O(1/\gamma)$; this implies that the number of eigenvectors with eigenvalue larger than $C\epsilon^2$ is at most $O(1/(\gamma\epsilon^2))$. Thus, if we take $\mathcal{V}$ to be the span of such eigenvectors, we would have that $\|\text{Proj}_{\mathcal{V}^\perp}(\mu)\| \leq \epsilon/2$ and $\dim(\mathcal{V}) = O(1/(\gamma\epsilon^2))$.

This procedure implements one round of our dimension reduction. For $\gamma = 1/\sqrt{d}$, this reduces the dimension from $d$ to $O(\sqrt{d}/\epsilon^2)$. That is, the dimension is reduced by a factor of 2 whenever $d$ is at least a large constant multiple of $1/\epsilon^4$. By repeating this procedure, we reduce the dimension down to $O(1/\epsilon^4)$. Then, applying the algorithm of Proposition 2.1 on the remaining subspace of dimension $d' = O(1/\epsilon^4)$, completes the algorithm using $2^{O(d'+1/\epsilon^2)} = 2^{O(1/\epsilon^4)}$ additional samples and time.

The first challenge towards implementing the above approach is approximating $A$ from samples. This is non-trivial, as $\exp(-(\|x\|^2 - d)\gamma)$ can be as large as $\exp(d\gamma)$, suggesting that we might need roughly this many samples. Fortunately, we note that $\|x\|^2$ is unlikely to be much less than $d$. In particular, the squared norm of a standard normal is approximately Gaussian distributed with mean $d$ and standard deviation $\sqrt{d}$. This implies that the expected size of $\exp(-(\|x\|^2 - d)\gamma)$ is approximately $\exp(d\gamma^2)$ (and it is smaller for Gaussians with other means). By simple concentration arguments, for each entry of the matrix $A$, it is not difficult to show that $\text{poly}(d/\epsilon)\exp(d\gamma^2)$ samples suf-

fice for $A$ to convergence . Thus, the choice $\gamma = 1/\sqrt{d}$ made in the previous paragraph suffices for our purposes (Lemma 2.8).

We emphasize that there are two problems with this approach to be addressed. The first is that the run-time of the algorithm presented so far is not polynomial: this is because the overall number of samples is $n = \text{poly}(d)2^{\Theta(1/\epsilon^2)}$, while the runtime of the final brute-force step is $\text{poly}(d)2^{\Theta(1/\epsilon^4)}$. Second, we would like the sample complexity to be linear in $d$ rather than $\text{poly}(d)$.

In order to resolve the first issue, we need our dimensionality reduction to be able to bring the dimension all the way down to $O(1/\epsilon^2)$— rather than $O(1/\epsilon^4)$. To achieve this, we can first reduce the dimension to $d' = O(1/\epsilon^4)$, as described in the earlier paragraphs, and then change the value of $\gamma$ to $1/(\sqrt{d'}\epsilon)$. This way, the next iteration will reduce $d'$ to $O(\sqrt{d'}/\epsilon)$ (which means that we can keep halving the dimension until $d'$ becomes a constant multiple of $1/\epsilon^2$). Using this value for $\gamma$ implies that each iteration will be using $\exp(O(d'\gamma^2)) = \exp(O(1/\epsilon^2))$ samples and time (Lemma 2.8). Finally, we apply Proposition 2.1 to the resulting subspace $\mathcal{V}$.

To improve the sample size dependence on the dimension (cf. Lemma 2.7), we require strong concentration bounds, in particular the Matrix Bernstein inequality. However, to apply this inequality, one needs a universal bound for our random variable $xx^\top \exp(-(\|x\|^2 - d)\gamma)$, which is bad when $\|x\|^2$ is small. Fortunately, this happens with small probability, and we are able to show that truncating the values when $\|x\|^2$ is much smaller than $d$ will not affect the final mean by much. However, taking $\gamma = 1/\sqrt{d}$ will prove not quite sufficient for our purposes: there will be roughly a $1/d$-fraction of samples with $\|x\|^2 = d - \Omega(\sqrt{d\log(d)})$, leading to terms with norm on the order of $d\exp(\Omega(\sqrt{\log(d)}))$, and it will take a slightly super-linear number of samples to average these away. To handle this, we need to use a slightly smaller value of $\gamma$, such as $1/\sqrt{d\log(d)}$.

**Notation** We use $\mathbb{Z}_+$ for the set of positive integers. We denote $[n] = \{1, \ldots, n\}$. For a vector $x$ we denote by $\|x\|$ its Euclidean norm. Let $I_d$ denote the $d \times d$ identity matrix (omitting the subscript when it is clear from the context). We use $\top$ for the transpose of matrices and vectors. For a subspace $\mathcal{V}$ of $\mathbb{R}^d$ of dimension $m$, we denote by $\Pi_{\mathcal{V}} \in \mathbb{R}^{d \times d}$ the orthogonal projection matrix of $\mathcal{V}$. That is, if the subspace $\mathcal{V}$ is spanned by the columns of the matrix $A$, then $\Pi_{\mathcal{V}} := A(A^\top A)^{-1}A^\top$. For a vector $x \in \mathbb{R}^d$, we use $\text{Proj}_{\mathcal{V}}(x) = \Pi_{\mathcal{V}}x$ to denote the orthogonal projection of $x$ onto $\mathcal{V}$. We say that a symmetric $d \times d$ matrix $A$ is PSD (positive semidefinite) and write $A \succeq 0$ if for all $x \in \mathbb{R}^d$ it holds $x^\top Ax \geq 0$. We use $\|A\|_{\text{op}}$ and $\|A\|_{\text{F}}$ for the operator (spectral) and Frobenius norm of a matrix $A$ respectively.

We write $x \sim D$ for a random variable $x$ following the distribution $D$ and use $\mathbf{E}[x]$ for its expectation. We use $\mathcal{N}(\mu, \Sigma)$ to denote the Gaussian distribution with mean $\mu$ and covariance matrix $\Sigma$. For a scalar random variable $x$, we define the $L_p$-norm of $x$ to be $\|x\|_{L_p} = \mathbf{E}[|x|^p]^{1/p}$. We use $a \lesssim b$ to denote that there exists an absolute universal constant $C > 0$ (independent of the variables or parameters on which $a$ and $b$ depend) such that $a \leq Cb$. We use $\text{polylog}()$ to denote a quantity that is polylogarithmic in its arguments and we use $\tilde{O}$ to hide such factors.

## 2. Efficient Robust Mean Estimation in the Mean-Shift Model: Proof of Theorem 1.2

In this section, we present our algorithm (Algorithm 1) and establish Theorem 1.2. Section 2.1 provides a sample-efficient but computationally inefficient estimator, which will be employed at the final step of our algorithm (after the dimension has been significantly decreased). Sections 2.2 and 2.3 analyze our dimension reduction procedure: Section 2.2 records the desired properties of the reweighted second moment matrix that our algorithm relies on, and shows that they hold with high probability with sufficiently many samples. Finally, Section 2.3 provides the core analysis of the dimension reduction procedure, and combines everything to prove Theorem 1.2.

### 2.1. Computationally Inefficient Multivariate Robust Estimator

Proposition 2.1 below provides a robust multivariate mean estimator for the mean-shift model that uses $n = d\,2^{O(1/\epsilon^2)}$ samples and $2^{O(d)}\text{poly}(n, d)$ runtime. Although the runtime is exponential in the dimension, it becomes just $\text{poly}(n, d)$ if $d = O(1/\epsilon^2)$. Therefore, this estimator will be useful after our dimension reduction technique that will be developed in the next sections manages to reduce the dimension to $O(1/\epsilon^2)$.

**Proposition 2.1** (Inefficient Estimator). *Let $d \in \mathbb{Z}_+$ denote the dimension, and $C$ be a sufficiently large absolute constant. Let $\alpha \leq 0.49$, $\epsilon > 0$, and $\delta \in (0, 1)$ be parameters, and $\mu \in \mathbb{R}^d$ be an (unknown) vector. There exists an algorithm that, on input $\epsilon$ and any set of $n \geq 2^{C/\epsilon^2}(d+\log(1/\delta))$ $\alpha$-corrupted set of points from $\mathcal{N}(\mu, I)$ under the mean-shift model (cf. Definition 1.1), outputs a $\widehat{\mu}$ such that $\|\widehat{\mu} - \mu\| \leq \epsilon$ with probability at least $1 - \delta$. Moreover, it runs in time $2^{O(d)}\text{poly}(n, d)$.*

Proposition 2.1 follows in a relatively standard way by taking a fine discretization of the unit sphere, running the one-dimensional estimator for each of the directions in that cover set, and combining the solutions to a vector (see, e.g., Section 1.5 of Diakonikolas & Kane (2023)). For completeness, we provide a proof of correctness in Appendix C using the

univariate estimator of Fact 2.2 as a black-box. A nice property of the prior work is that the robust univariate estimator has breakdown point arbitrarily close to $1/2$ and does not need to know the contamination parameter (c.f. Section 5.3 of Kotekal & Gao (2025)).

**Fact 2.2** (One-dimensional estimator, see, e.g., Li (2023); Kotekal & Gao (2025)). *Let $\mu \in \mathbb{R}$ be an (unknown) mean. Let $C$ be a sufficiently large constant and $\alpha \leq 0.49$. There is an algorithm that given, $\epsilon > 0$, $\delta \in (0,1)$ and a set of $n = 2^{C/\epsilon^2} \log(1/\delta)$ $\alpha$-corrupted samples from $\mathcal{N}(\mu, 1)$ according to the mean-shift model (Definition 1.1), finds $\widehat{\mu} \in \mathbb{R}$ such that, with probability at least $1 - \delta$, it holds $|\widehat{\mu} - \mu| \leq \epsilon$. The runtime of the algorithm is $\mathrm{poly}(n)$.*

## 2.2. Reweighted Second Moment Matrix

The structure of this section is as follows. Equation (1) defines the reweighted second moment matrix that will be used by our dimensionality reduction procedure. Subsequently, Definition 2.6 states the deterministic conditions regarding this matrix that will be required for the correctness of our dimension reduction algorithm. Finally, in Lemmas 2.7 and 2.8, we show that the matrix satisfies these conditions given sufficiently many samples.

For $\beta \in (0,1]$, a subspace $\mathcal{V} \subseteq \mathbb{R}^d$ of dimension $k \leq d$, and points $x_1, \ldots, x_n \in \mathcal{V}$, define the matrix:

$$\widehat{A} := \frac{1}{n} \sum_{i=1}^{n} \widehat{A}_i, \quad \text{where} \quad \widehat{A}_i := F_{\beta,k}(x_i) Z_{\beta,k},$$

$$Z_{\beta,k} := \left(1 + \frac{2}{\beta\sqrt{k}}\right)^{\frac{k}{2}+2},$$

$$F_{\beta,k}(x) := \left(xx^\top - \frac{\beta\sqrt{k}}{\beta\sqrt{k}+2}\Pi_\mathcal{V}\right) e^{-\frac{\|x\|^2}{\beta\sqrt{k}}}, \quad (1)$$

where $\Pi_\mathcal{V}$ denotes the orthogonal projection matrix of the subspace $\mathcal{V}$. The subspace $\mathcal{V}$ will be the one that the algorithm maintains in each round (and whose dimension decreases in every round). For simplicity, the reader can think of $\mathcal{V} = \mathbb{R}^k$ and $\Pi_\mathcal{V} = I_k$; however, since the basis of the subspace may not be aligned with the elements of the standard orthonormal basis, using $\Pi_\mathcal{V}$ is required in general.

The definition of $\widehat{A}$ has been designed so that the expectation of a single sample's deviation, namely $(xx^\top - \frac{\beta\sqrt{k}}{\beta\sqrt{k}+2}I)e^{-\frac{\|x\|^2}{\beta\sqrt{k}}}$ when $x \sim \mathcal{N}(z, I)$ is roughly proportional to $zz^\top e^{-\frac{\|z\|^2}{\beta\sqrt{k}}}$. This is shown in Lemma 2.3 below. The factor $Z_{\beta,k}$ is the appropriate normalization factor that arises.

**Lemma 2.3.** *Let $\mathcal{V} \subseteq \mathbb{R}^d$ be a subspace of $\mathbb{R}^d$ of dimension $k \leq d$. For any $z \in \mathcal{V}$, we have that $\mathbf{E}_{x \sim \mathcal{N}(z, \Pi_\mathcal{V})}[F_{\beta,k}(x)] Z_{\beta,k} = zz^\top e^{-\frac{\|z\|^2}{\beta\sqrt{k}+2}}$, where $F_{\beta,k}$ and $Z_{\beta,k}$ are as defined in Equation (1).*

---

**Algorithm 1** Mean Estimation under Definition 1.1

1: **Input**: Accuracy $\epsilon > 0$.

2: **Output**: $\widehat{\mu} \in \mathbb{R}^d$ such that $\|\widehat{\mu} - \mu\| \leq \epsilon$.

3: Fix $C$ a sufficiently large constant, $n_0 = Cd$, $n_1 = (d/\epsilon^{2+o(1)}) \log^C(d)$, and $n_2 = 2^{C/\epsilon^2} \log^C(d)$.

4: /*Rough Estimation:*/

5: Draw $T_0$, a set of $n_0$ corrupted points according to Definition 1.1.

6: Use $T_0$ to find $\widehat{\mu}_0$ with $\|\widehat{\mu}_0 - \mu\| = O(1)$. {e.g., Corollary 2.12 and Exercise 2.10 in Diakonikolas & Kane (2023)}

7: /*Dimension Reduction:*/

8: Initialize $t \leftarrow 1$, $k \leftarrow d$ and $\mathcal{V}_1 = \mathbb{R}^k$. {$k$ will denote the dimension of the current subspace}

9: **while** $k \geq 1/\epsilon^2$ **do**

10:     **if** $k \geq C \log^4(d)/\epsilon^5$ **then**

11:         set $\beta \leftarrow \sqrt{\log(k)}$ and $N \leftarrow n_1$, $\eta_t \leftarrow (\epsilon/\log d)^2$.

12:     **else**

13:         set $\beta \leftarrow \epsilon$ and $N \leftarrow n_2$, $\eta_t \leftarrow 36\epsilon/\sqrt{k}$.

14:     **end if**

15:     Draw a set $T'_t$ of $N$ corrupted points from the model of Definition 1.1.

16:     $T_t \leftarrow \{\mathrm{Proj}_{\mathcal{V}_t}(x - \widehat{\mu}_0) : x \in T'_t\}$.

17:     $\widehat{A}_t \leftarrow \frac{1}{|T_t|} \sum_{x_i \in T_t} \widehat{A}_i$, as defined in (1).

18:     Find the eigenvectors of $v_1^{(t+1)}, \ldots, v_{k'}^{(t+1)}$ of $\widehat{A}_t$ with eigenvalue at least $\eta_t$ (let $k'$ denote the number of such eigenvectors).

19:     Let $\mathcal{V}_t$ be the subspace spanned by $\{v_1^{(t+1)}, \ldots, v_{k'}^{(t+1)}\}$.

20:     Update $k \leftarrow k'$ and $t \leftarrow t + 1$.

21: **end while**

22: /*Run Inefficient Algorithm:*/

23: Sample a set $T'_t$ of $n_2$ corrupted points from the model of Definition 1.1.

24: $T_t \leftarrow \{\mathrm{Proj}_{\mathcal{V}_t}(x) : x \in T'_t\}$.

25: Use $T_t$ to find $\widehat{\mu}_1 \in \mathcal{V}_t$ with $\|\widehat{\mu}_1 - \mathrm{Proj}_{\mathcal{V}_t}(\mu)\| \leq \epsilon$. {Use Algorithm from Proposition 2.1}

26: **return** $\mathrm{Proj}_{\mathcal{V}_t^\perp}(\widehat{\mu}_0) + \widehat{\mu}_1$.

---

The deterministic conditions that we require to hold with high probability quantify how close $\widehat{A}$ needs to be to its expectation $A := \frac{1}{n}\sum_{i=1}^{n} z_i z_i^\top e^{-\frac{\|z_i\|^2}{\beta\sqrt{k}+2}}$. For the inlier points, we will require closeness of the reweighted second moment to its expectation in operator norm (cf. Definition 2.4). For the outlier samples, it will suffice to have the milder condition that the reweighted second moment is not too negative (cf. Definition 2.5). Finally, note that the expectation matrix $A := \frac{1}{n}\sum_{i=1}^{n} z_i z_i^\top e^{-\frac{\|z_i\|^2}{\beta\sqrt{k}+2}}$ has $\operatorname{tr}(A) = \frac{1}{n}\sum_{i=1}^{n} \|z_i\|^2 e^{-\frac{\|z_i\|^2}{\beta\sqrt{k}+2}} \le \beta\sqrt{k} + 2$, where we used the elementary inequality $ye^{-y/\gamma} \le \gamma$ for $y > 0$. The bounded trace will be a key property for the correctness of our dimensionality reduction procedure. We will thus require the same property to hold for the empirical reweighted moment matrix. Definition 2.6 combines all three of the aforementioned conditions.

**Definition 2.4** (($\eta, \beta$)-concentrated set)**.** *Let $\eta, \beta > 0$ be parameters, $\mathcal{V}$ be a $k$-dimensional subspace of $\mathbb{R}^d$ and let $x_1, \ldots, x_n \in \mathcal{V}$. We say that the set $\{x_1, \ldots, x_n\}$ is $(\eta, \beta)$-concentrated with respect to $\mathcal{V}$ and $z_1, \ldots, z_n \in \mathcal{V}$, if the matrix $\widehat{A}$ defined in Equation (1) satisfies $\|\widehat{A} - A\|_{\mathrm{op}} \le \eta$, where $A := \frac{1}{n}\sum_{i=1}^{n} z_i z_i^\top e^{-\frac{\|z_i\|^2}{\beta\sqrt{k}+2}}$.*

**Definition 2.5** (($\eta, \beta$)-positive definite)**.** *Let $\eta, \beta > 0$ be parameters and $\mathcal{V}$ be a $k$-dimensional subspace of $\mathbb{R}^d$. We say that the set of points $x_1, \ldots, x_n \in \mathcal{V}$ is $(\eta, \beta)$-positive definite, if the matrix $\widehat{A}$ defined in Equation (1) satisfies $v^\top \widehat{A} v \ge -\eta$, for all $v \in \mathcal{V}$ with $\|v\| = 1$.*

**Definition 2.6** (($\eta, \beta$)-good set)**.** *Let $\eta, \beta > 0$ be parameters and $\mathcal{V}$ be a $k$-dimensional subspace of $\mathbb{R}^d$. A set $T$ of points in $\mathcal{V}$ is called $(\eta, \beta)$-good with respect to $\mathcal{V}$ and the vectors $\mu, z_1, \ldots, z_{\alpha n} \in \mathcal{V}$ if there exists $S \subseteq T$, with $|S| = (1-\alpha)|T|$ such that:*

1. *(Condition for inliers) $S$ is $(\eta, \beta)$-concentrated with respect to $\mathcal{V}$ and the vector sequence $\mu, \ldots, \mu$ (i.e., the sequence that has the $\mu$ vector $(1 - \alpha)n$ times).*

2. *(Condition for outliers) $T \setminus S$ is $(\eta, \beta)$-positive definite.*

3. *(Bounded trace condition) The matrix $\widehat{A}$ from (1) computed over all $x_i \in T$ has $\operatorname{tr}(\widehat{A}) \le 18\beta\sqrt{k}$.*

Lemmas 2.7 and 2.8 show that a sufficiently large set of samples from our model satisfies our $(\eta, \beta)$-goodness conditions with high probability. We need two lemmata because our algorithm will use $\beta = \sqrt{\log k}$ for the most part, and $\beta = \epsilon$ for the last few iterations.

For the case $\beta = \sqrt{\log k}$ (corresponding to Lemma 2.7), we need to use strong concentration bounds in order for the sample complexity to scale linearly (up to polylog factors) with the dimension $k$ (see proof sketch at the end of this section and full proof in Appendix D).

**Lemma 2.7.** *Let $\eta \in (0, 1)$ and $k \in \mathbb{Z}_+$ denote the dimension, and assume $k$ is bigger than a sufficiently large constant. There exists sample size $n = k\log^3(k)(1/\eta)^{2+o(1)}\frac{1}{\delta}$ such that the following holds: Let $T$ be a set of $n$ $\alpha$-corrupted points from $\mathcal{N}(\mu, I)$ according to Definition 1.1 with the assumption that $\|\mu\| = O(1)$. Denote by $z_1, \ldots, z_{\alpha n}$ the adversarial centers in Definition 1.1. Then, with probability at least $1 - \delta$, $T$ is $(\eta, \sqrt{\log k})$-good with respect to $\mu, z_1, \ldots, z_{\alpha n}$.*

For the case $\beta = \epsilon$, we can resort to a simpler proof which consists of calculating the variance of each entry of our random matrix and using Chebyshev's inequality entry-wise. This is done in Lemma 2.8. Although the resulting dependence on the dimension $k$ scales as $k^5$, we will use this bound only after the dimension has decreased to roughly $1/\epsilon^5$; meaning that the sample complexity will be dominated by the factor $2^{C/\epsilon^2}$. The full proof can also be found in Appendix D.

**Lemma 2.8.** *Let $T$ be a set of $n$ $\alpha$-corrupted set of points from $\mathcal{N}(\mu, I)$ according to Definition 1.1 for some $\mu$ with $\|\mu\| = O(1)$. Assume $n = \frac{k^5}{\eta^2\delta}2^{C/\epsilon^2}$ for a sufficiently large absolute constant $C$, where $k \ge 1/\epsilon^2$ and $\eta \le \epsilon$. Denote by $z_1, \ldots, z_{\alpha n}$ the adversarial points used in Definition 1.1. Then, with probability at least $1 - \delta$, $T$ is $(\eta, \epsilon)$-good with respect to $\mu, z_1, \ldots, z_{\alpha n}$.*

We conclude this section with a brief proof sketch of the first of these lemmata.

*Proof Sketch of Lemma 2.7.* We focus on showing Item 2 ($\eta$-positive definiteness) for the set of outliers, as the other parts can be proved similarly. Let $\widehat{A} = \sum_{i:x_i \in T \setminus S} \widehat{A}_i$, where $\widehat{A}_i$ as defined in Equation (1). First note that, using Lemma 2.3, we have that $\mathbf{E}[\widehat{A}] = A := \frac{1}{\alpha n}\sum_i z_i z_i^\top e^{-\frac{\|z_i\|^2}{\sqrt{k}\beta+2}}$ is positive semidefinite. Hence, in order to prove that $\widehat{A}$ is $\eta$-positive definite, it suffices to prove that $\widehat{A}$ cannot be much smaller than $A$ in every direction.

First note that the $e^{-\frac{\|x_i\|^2}{\sqrt{k}\beta}}Z_{\beta,k}$ factor appearing in the definition of $\widehat{A}$ behaves roughly like $e^{-\frac{\|x_i\|^2-k}{\sqrt{k}\log k}}$, as $Z_{\beta,k} = \Theta(e^{\sqrt{k/\log k}})$ for $\beta = \sqrt{\log k}$. Towards showing concentration with a linear number of samples, we first note that the $\|x_i\|^2 - k$ (appearing before) is bounded, except with tiny probability. That is, we argue that, without loss of generality, we can work with the matrix of the form $\widehat{A}' = \sum_{x_i \in T \setminus S} \widehat{A}_i \mathbb{1}(\mathcal{E}_i)$ instead of $\widehat{A}$. Here $\mathcal{E}_i$ is defined to be the following *good event*: $|\|x_i\|^2 - (\|z_i\|^2 + k)| \lesssim \log(1/\tau) + (\sqrt{k} + \|z_i\|)\sqrt{\log(1/\tau)}$, where $\tau := (1/n)^4$. This is indeed without loss of generality because: (i) $\mathcal{E}_i$ hold with probability $1 - \delta$ for all $i$ simultaneously (each $\mathcal{E}_i$ fails

with probability $\tau$ by Fact B.5, and thus the probability that there exists $\mathcal{E}_i$ that fails is at most $n\tau \leq 1/n^3 \leq \delta$ by union bound); and (ii) we can prove that $\|\mathbb{E}[\widehat{A}'] - A\| \leq \eta/2$, i.e., the truncation shifts the mean by a small amount.

Then we can apply standard concentration results on $\widehat{A}'$, as our expressions are appropriately bounded. We achieve this by carefully decomposing $\widehat{A}'$ into three terms by writing $x_i = z_i + g_i$, where $g_i \sim \mathcal{N}(0, I)$: $\widehat{A}_i = ((g_i g_i^\top - \frac{\sqrt{k \log(k)}}{\sqrt{k \log(k)}+2}I) + (z_i g_i^\top + g_i z_i^\top) + z_i z_i^\top)e^{-\frac{\|x_i\|^2}{\sqrt{k \log(k)}}}Z_{\beta,k} =: \widehat{T}_{1i} + \widehat{T}_{2i} + \widehat{T}_{3i}$. For the first and second term, we use Hoeffding's inequality for subgaussian random variables along with a cover argument. For the third term, we use the Matrix Bernstein Fact B.4 inequality. This application requires an absolute bound on $\|T_{3i}\| \leq K$ that holds almost surely, as well as a bound for the matrix variance $\|\mathbb{E}[\widehat{T}_{3i}^2]\|$. Given the "good events" from above, we can use that $\exp(-\|x\|^2/\sqrt{k \log k})Z_{\sqrt{\log k},k}$ is roughly less than $\exp(-\|z_i\|^2/\sqrt{k \log k})$. We can thus obtain that $\|T_{3i}\| \leq \|z_i\|^2 \exp(-\|z_i\|^2/\sqrt{k \log k}) \lesssim \sqrt{k \log k}$ and $\|z_i\|^4 \exp(-2\|z_i\|^2/\sqrt{k \log k}) \lesssim k \log k$, which results in near-linear in $k$ sample complexity. $\qquad\square$

## 2.3. Analysis of Algorithm 1

The lemma below is the core of the analysis for a single iteration of our dimensionality reduction technique. Since the sample version $\widehat{A}_t$ of the matrix, that we use in Algorithm 1 of Algorithm 1, is close to its expectation, we show that our naïve estimator from the first step of the algorithm is accurate inside the subspace of small eigenvalues of $\widehat{A}_t$ (see Lemma 2.9). Combined with Item 3 of Definition 2.6, which bounds from above the number of high eigenvalues, we conclude that a single round of our dimensionality reduction loop halves the dimension without accumulating error.

**Lemma 2.9.** *Let $\beta > 0$ and $\mathcal{V}$ be a subspace of $\mathbb{R}^d$ of dimension $k \leq d$. Let $T$ be an $(\eta, \beta)$-good set of $n$ samples (cf. Definition 2.6) with respect to the subspace $\mathcal{V}$ and the vectors $\mu, z_1, \ldots, z_{\alpha n} \in \mathcal{V}$, and assume that $\|\mu\| = O(1)$. Define $\widehat{A}$ as in (1). If $\mathcal{U}$ is a subspace of $\mathcal{V}$ such that $v^\top \widehat{A} v \leq \eta$ for all unit vectors $v \in \mathcal{U}$, then we have that $\|\mathrm{Proj}_{\mathcal{U}}(\mu)\| = O(\sqrt{\eta})$, where $\mathrm{Proj}_{\mathcal{U}}(\cdot)$ denotes the projection to subspace $\mathcal{U}$.*

*Proof.* Without loss of generality, we will prove the lemma for the case where $\mathcal{V} = \mathbb{R}^k$. Since $T$ is $(\eta, \beta)$-good, it can be partitioned into $S$ (inliers) and $T \setminus S$ (outliers), where the two sets satisfy the properties of Definition 2.6. Consider the matrices

$$\widehat{A}_{\text{inliers}} = \frac{Z_{\beta,k}}{n} \sum_{x \in S} F_{\beta,k}(x),$$

$$A_{\text{inliers}} = \mu\mu^\top e^{-\frac{\|\mu\|^2}{\beta\sqrt{k}+2}},$$

$$\widehat{A}_{\text{outliers}} = \frac{Z_{\beta,k}}{\alpha n} \sum_{x \in T \setminus S} F_{\beta,k}(x),$$

$$A_{\text{outliers}} = \frac{1}{\alpha n} \sum_{i:x_i \in T \setminus S} z_i z_i^\top e^{-\frac{\|z_i\|^2}{\beta\sqrt{k}+2}},$$

where $Z_{\beta,k}$ and $F_{\beta,k}$ are as defined in Equation (1). We can decompose $\widehat{A}$ into inliers and outliers, i.e., $\widehat{A} = (1-\alpha)\widehat{A}_{\text{inliers}} + \alpha\widehat{A}_{\text{outliers}}$. By the assumption that $T$ is $(\eta, \beta)$-good (cf. Definition 2.6), we have that $T \setminus S$ is $\eta$-positive definite (cf. Definition 2.5); hence $v^\top \widehat{A}_{\text{outliers}} v \geq -\eta$.

By assumption, $S$ is $\eta$-concentrated (Item 1 of Definition 2.6); thus $|v^\top \widehat{A}_{\text{inliers}} v - v^\top A_{\text{inliers}} v| \leq \eta$.

Putting everything together, for every unit vector $v$ for which $v^\top \widehat{A} v \leq \eta$, we have that:

$$(1-\alpha)v^\top A_{\text{inliers}} v \qquad\qquad (2)$$
$$= (1-\alpha)v^\top \widehat{A}_{\text{inliers}} v + (1-\alpha)v^\top (A_{\text{inliers}} - \widehat{A}_{\text{inliers}})v$$
$$= v^\top \widehat{A} v - \alpha v^\top \widehat{A}_{\text{outliers}} v + (1-\alpha)v^\top (A_{\text{inliers}} - \widehat{A}_{\text{inliers}})v$$
$$\leq \eta + \alpha\eta + (1-\alpha)\eta \leq 2\eta .$$

Using that $A_{\text{inliers}} = \mu\mu^\top e^{-\frac{\|\mu\|^2}{\beta\sqrt{k}+2}}$, the above implies that $(v^\top\mu)^2 \leq 2\eta e^{\frac{\|\mu\|^2}{\beta\sqrt{k}+2}} \lesssim \eta$, where the last inequality is because $\|\mu\| = O(1)$. Finally, since $\|\mathrm{Proj}_{\mathcal{U}}(\mu)\| = \max_{v \in \mathcal{U}:\|v\|=1}|v^\top\mu|$, it follows that $\|\mathrm{Proj}_{\mathcal{U}}(\mu)\| \leq O(\sqrt{\eta})$. $\qquad\square$

### 2.3.1. PROOF OF THEOREM 1.2

We are now ready to prove our main theorem. The full proof is deferred to Appendix E. We use the same notation as in the pseudocode provided in Algorithm 1: the $t$-th iteration of the while loop maintains a subspace $\mathcal{V}_t$, whose dimension, $k$, starts from $d$ and can only decrease from a round to the next one. This while loop has two distinct phases: Phase 1 will refer to all the iterations during which $C \log^4(d)/\epsilon^5 \leq k \leq d$, and Phase 2 will refer to all the iterations with $1/\epsilon^2 \leq k < C \log^4(d)/\epsilon^5$. The analysis of the algorithm consists of the claims stated below. The claims are that each of the following holds with high constant probability:

1. **Warm start**: If $\widehat{\mu}_0$ is the estimator from line 1 of Algorithm 1, then $\|\widehat{\mu}_0 - \mu\| = O(1)$.

2. **Dimension Reduction**: If $T_1, T_2$ denote the number of iterations of Phase 1 and Phase 2 respectively, then $T_1 \leq \log(d)$, $T_2 \leq 100\log(\log(d)/\epsilon)$. Moreover, for all $t = 1, \ldots, T_1 + T_2$, we have $\|\mathrm{Proj}_{\mathcal{V}_{t+1}^\perp}(\widehat{\mu}_0 - \mu)\| \lesssim \sum_{t'=1}^t \sqrt{\eta_{t'}}$, where $\eta_{t'}$ are the values set in lines 1 and 1.

3. **Estimator for the remaining subspace**: The lines 1-1 of Algorithm 1 find a vector $\widehat{\mu}_1 \in \mathcal{V}_t$ such that $\|\widehat{\mu}_1 - \mathrm{Proj}_{\mathcal{V}_t}(\mu)\| \leq \epsilon$.

Here we will sketch how each claim can be proved, with the full details in Appendix E. We start by showing how the claims imply that $\|\widehat{\mu} - \mu\| = O(\epsilon)$ for $\widehat{\mu} := \widehat{\mu}_0 + \widehat{\mu}_1$.[2]

Consider $t = T_1 + T_2$, so that $\mathcal{V}_t$ denotes the subspace after exiting the while loop. By decomposing the true mean into the projections onto the two orthogonal subspaces we have that $\mu = \mathrm{Proj}_{\mathcal{V}_t^\perp}(\mu) + \mathrm{Proj}_{\mathcal{V}_t}(\mu)$. By the Pythagorean theorem, we can write

$$\|\widehat{\mu} - \mu\|^2 = \|\mathrm{Proj}_{\mathcal{V}_t^\perp}(\widehat{\mu}_0 - \mu) + \mathrm{Proj}_{\mathcal{V}_t}(\widehat{\mu}_1 - \mu)\|^2$$
$$= \|\mathrm{Proj}_{\mathcal{V}_t^\perp}(\widehat{\mu}_0 - \mu)\|^2 + \|\mathrm{Proj}_{\mathcal{V}_t}(\widehat{\mu}_1 - \mu)\|^2.$$

The last term is $\|\mathrm{Proj}_{\mathcal{V}_t}(\widehat{\mu}_1 - \mu)\| \leq \epsilon$ by Item 3. It suffices to bound the first term by $O(\epsilon)$. Towards this end, denote $D := C \log^4(d)/\epsilon^5$, which is the dimension during the first iteration of Phase 2. Then,

$$\|\mathrm{Proj}_{\mathcal{V}_t^\perp}(\widehat{\mu}_0 - \mu)\|$$
$$\lesssim \sum_{t'=1}^{T_1+T_2} \sqrt{\eta_{t'}} = \sum_{t'=1}^{T_1} \sqrt{\eta_{t'}} + \sum_{t'=T_1+1}^{T_1+T_2} \sqrt{\eta_{t'}}$$
$$\text{(from Item 2)}$$
$$\leq T_1 \frac{\epsilon}{\log d} + \sum_{t'=T_1+1}^{T_1+T_2} \sqrt{\eta_{t'}} \leq \epsilon + \sum_{t'=T_1+1}^{T_1+T_2} \sqrt{\eta_{t'}}$$
$$\leq \epsilon + \left( \sqrt{\frac{36\epsilon}{\sqrt{D}}} + \sqrt{\frac{36\epsilon}{\sqrt{D/2}}} + \cdots + \sqrt{\frac{36\epsilon}{\sqrt{1/\epsilon^2}}} \right)$$
$$\leq \epsilon + \frac{6\sqrt{\epsilon}}{D^{1/4}} \sum_{i=0}^{\lg(D\epsilon^2)} 2^{i/4} \leq \epsilon + \frac{6\sqrt{\epsilon}}{(2^{1/4}-1)D^{1/4}} 2^{\frac{\lg(D\epsilon^2)}{4}}$$
$$= \epsilon + \frac{6\sqrt{\epsilon}}{(2^{1/4}-1)D^{1/4}}(D\epsilon^2)^{1/4} \lesssim \epsilon,$$

where we used the definition of $\eta_{t'}$ from lines 1,1, and direct calculations for the series ($\lg(\cdot)$ denotes the logarithm with base 2).

Finally, we discuss briefly the proofs of the claims in Items 1 to 3. The first and the last follow immediately by Diakonikolas & Kane (2023) (Corollary 2.12 and Exercise 2.10) and Proposition 2.1 respectively. Regarding Item 2, let us first consider Phase 1 (iterations during which $k \geq C \log^4(d)/\epsilon^5$). By an application of Lemma 2.7 with $\eta = \eta_t = (\epsilon/\log d)^2$ and a union bound, all the sets $T_t$ drawn in line 1 will be $(\eta_t, \sqrt{\log k})$-good with high constant

---

[2]The guarantee of Theorem 1.2 is $\|\widehat{\mu} - \mu\| \leq \epsilon$. Here we show $O(\epsilon)$ in order to keep the constants that appear simple. This is w.l.o.g. as one can later replace $\epsilon$ by $\epsilon/C$ for a large enough constant $C$ to obtain Theorem 1.2.

probability. Note that because of the warm start in line 1 and the transformation subtracting $\widehat{\mu}_0$ in Algorithm 1, every sample essentially comes from the mean-shift with mean $O(1)$, which makes Lemma 2.7 applicable. The number of eigenvectors with eigenvalue larger than $\eta_t$ is at most $\mathrm{tr}(\widehat{A}_t)/\eta_t \leq 18\sqrt{k \log k}/\eta_t = O((\log d)^2 \epsilon^{-2}\sqrt{\log k}) \leq O(\log^{2.5}(d)\epsilon^{-2})$, where we first used the definition of $(\eta_t, \sqrt{\log k})$-goodness (Definition 2.6) and then that $\eta_t = (\epsilon/\log d)^2$. Thus, as long as $k > C \log^5(d)/\epsilon^4$, the dimension during each round gets halved. After $T_1 = O(\log d)$ such iterations, the dimension reaches $C \log^5(d)/\epsilon^4$, after which the aforementioned no longer guarantees that it will continue to drop. This is the reason why we need to change the value of $\beta$ to $\epsilon$, and set $\eta_t := 36\epsilon/\sqrt{k}$ in Algorithm 1 for the remaining iterations (Phase 2). The argument for Phase 2 is similar, but uses different parameters: our datasets are now $(\eta_t, \epsilon)$-good by Lemma 2.8, and the number of eigenvectors larger than $\eta_t$ is now at most $\mathrm{tr}(\widehat{A}_t)/\eta_t \leq 18\beta\sqrt{k}/\eta_t = k/2$. Thus, Phase 2 will continue to halve the dimension. Finally, the claim that $\|\mathrm{Proj}_{\mathcal{V}_{t+1}^\perp}(\widehat{\mu}_0 - \mu)\| \lesssim \sum_{t'=1}^{t} \sqrt{\eta_{t'}}$ follows immediately from Lemma 2.9.

**Runtime and Sample Complexity of Algorithm 1** It can be readily verified that the number of samples $n_0, n_1, n_2$ defined in Algorithm 1 of the algorithm suffice for the aforementioned applications of Lemmas 2.7 and 2.8. Thus, the overall sample complexity of the algorithm is $n = n_0 + n_1 \cdot T_1 + n_2 \cdot (T_2 + 1) = O(d) + d \, \mathrm{polylog}(d)\epsilon^{-(2+o(1))} + 2^{O(1/\epsilon^2)} \mathrm{polylog}(d)$. Regarding runtime, the most computationally intensive part is the last step of the algorithm (Algorithm 1), which has runtime $\tau = 2^{O(k)} \mathrm{poly}(n_2, d)$, where the $k = 1/\epsilon^2$ here denotes the dimension of the final subspace. Since $n_2 = 2^{\Theta(1/\epsilon^2)} \mathrm{polylog}(d)$, that runtime is sample-polynomial, i.e., $\mathrm{poly}(n_2, d)$. It is easy to check that the other parts of the algorithm are also polynomial in the size of the input.

This completes the proof of Theorem 1.2. □

## 3. Conclusions and Open Problems

In this paper, we provide the first polynomial-time algorithm for high-dimensional mean estimation under mean-shift outliers. In this model, the outliers are not completely adversarial as they include a randomized component. This additional structure allows for consistent estimation, which is unattainable in purely adversarial models.

Our work takes a first algorithmic step in understanding the complexity of high-dimensional estimation for the known covariance mean-shift model of Definition 1.1. A number of concrete open problems and future directions suggest themselves. What is the complexity of robust mean estimation for the known covariance case when the underlying distribution

is not Gaussian? Is there a computationally efficient learner for the unknown covariance Gaussian case? We hope that this work will inspire further algorithmic work in robust estimation under structured contamination models.

## Acknowledgments

ID was supported by NSF Medium Award CCF-2107079 and an H.I. Romnes Faculty Fellowship. GI was supported in part by NSF Medium Award CCF-2107079 and NSF Award DMS-2023239 (TRIPODS). DK was supported by NSF Medium Award CCF-2107547 and NSF Award CCF-1553288 (CAREER). TP was supported by NSF Medium Award CCF-2107079.

## Impact Statement

The primary contribution of this research lies in theoretical advancements and foundational insights. Therefore, we believe that it does not have direct implications for societal, ethical, or policy-related issues.

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

## Supplemental Material

The appendix is structured as follows: First, Appendix A includes a more detailed summary of related work. Appendix B includes additional preliminaries required in subsequent technical sections. Appendix C gives the correctness analysis of Proposition 2.1. Appendix D contains the proofs of the technical lemmas involving the reweighted matrix used in Section 2.2. Appendix E contains the full proof of our main result (Theorem 1.2). Finally, Appendix F establishes Theorem F.1, which demonstrates adaptivity to unknown contamination parameter $\alpha$ arbitrarily close to $1/2$.

## A. Additional Related Work

**Additional Related Work on Mean Shift Contamination**   The most closely related work to ours is Kotekal & Gao (2025). They study the same problem as ours in one dimension and derive matching information theoretic upper and lower bounds for mean estimation. Kotekal & Gao (2025) also considers estimating the variance in the case when it is unknown to the algorithm. A similar upper bound for mean estimation in the unit variance case was also given in Li (2023). In earlier work, Carpentier et al. (2021) studied the sample complexity of robust mean estimation for the special case of Definition 1.1 in one dimension where $z_i - \mu > 0$.

More broadly, the problem of mean estimation with mean shift outliers has its roots in influential work by Bradley Efron (Efron, 2004; 2007; 2008) in the context of multiple hypothesis testing. In these works, Efron noticed through empirical evidence that, because the parameters of the null distribution are unknown, testing should be done in two stages: (i) estimation of the null parameters, and (ii) testing of null vs alternative hypothesis using standard multiple testing procedures. Although the focus of our paper theoretical, we refer to the discussions in Carpentier et al. (2021); Kotekal & Gao (2025), the CIRM talk in Gao (2024) and the references therein, for a discussion of the connection between the mean-shift noise model and Efron's work.

**Variance Parameter**   We highlight two points regarding the variance of samples in the mean-shift model. First, the variance may be unknown to the algorithm, unlike the identity/known covariance case in Definition 1.1. Specifically, consider a variant of the model (in one dimension for simplicity) where inliers are drawn from $\mathcal{N}(\mu, \sigma^2)$ and outliers from $\mathcal{N}(z_i, \sigma^2)$. This setting has been studied in prior work (see, e.g., Cai & Jin (2010); Carpentier et al. (2021); Kotekal & Gao (2025)). Notably, the last two works first analyze the case where the variance is known and equal to one, and then extend to the more general setting where it is unknown, designing estimators for both the variance and the mean. Even when the goal is solely mean estimation, their approach requires variance estimation as a first step. In contrast, Li (2023) considers only the known variance case.

The second point concerns what happens to the optimal error if outliers have different variances than inliers. Concretely, consider the model where inliers are drawn from $\mathcal{N}(\mu, \sigma^2)$ and outliers from $\mathcal{N}(z_i, \sigma_i^2)$. If outliers can have smaller variance than the inliers ($\sigma_i < \sigma$), then estimation is intrinsically harder than in the case where $\sigma_i \geq \sigma$. The reason is that estimation under Huber contamination can be reduced to estimation in the mean-shift model with $\sigma_i < \sigma$. Specifically consider the Huber contaminated sample with inliers that follow $\mathcal{N}(\mu, \sigma^2)$ and outliers $z_i$. By adding $\mathcal{N}(0, \alpha^2)$ noise (with $\alpha > 0$) to both inliers and outliers, we have that the inliers after the transformation follow $\mathcal{N}(\mu, \sigma^2 + \alpha^2)$ and the outliers follow $\mathcal{N}(z_i, \alpha^2)$, which is an instance of mean-shift contamination with $\sigma_i < \sigma$. Importantly, the consequence is that estimation with error that goes to zero is not possible in the mean-shift model where outliers have smaller variance than the inliers. There exists experimental work in the literature (Cai & Sun, 2009) which includes experiments with both smaller and larger variance, however as we mentioned provable consistent estimation is impossible in such cases.

Furthermore, note that in the setting where $\sigma_i \geq \sigma$, we can essentially assume that all $\sigma_i$ are exactly equal to $\sigma$. To see why, consider the random variable for a single outlier sample: $x_i = z_i + \mathcal{N}(0, \sigma_i^2)$. Observe that we can rewrite $x_i$ as $x_i = (z_i + \mathcal{N}(0, \sigma_i^2 - \sigma^2)) + \mathcal{N}(0, \sigma^2)$. Hence, by treating the outlier centers themselves as random variables, one can run an algorithm that solves the mean-estimation problem in the mean-shift model with unknown variance. Finally, the same argument applies in higher dimensions, in the setting where $\Sigma_i \succeq \Sigma$. Let $\Sigma, \Sigma_i$ be PSD matrices with $\Sigma_i \succeq \Sigma$. Then we can rewrite an outlier sample of the form $x_i = z_i + \mathcal{N}(0, \Sigma_i)$ as $x_i = (z_i + \mathcal{N}(0, \Sigma_i - \Sigma)) + \mathcal{N}(0, \Sigma)$, since $\Sigma_i - \Sigma$ is PSD and therefore a valid covariance matrix.

**Mean Shift Models for Other Problems**   The concept of more structured corruptions in the form of mean shifts has also been studied in the regression setting (Sardy et al., 2001; Gannaz, 2007; McCann & Welsch, 2007; She & Owen, 2011). In this model, it is assumed that $y_i = \beta^\top x_i + \gamma_i + \xi_i$, where $\gamma_i$ are adversarial mean shifts on top of the standard Gaussian

additive noise $\xi_i \sim \mathcal{N}(0, \sigma^2)$.

**Comparison with Lai et al. (2016)**    The approach of dimension reduction until a low-dimensional inefficient estimator can be employed has also appeared in previous robust statistics work (Lai et al., 2016), however that similarity is rather superficial and the underlying techniques are significantly different. Each iteration of dimension reduction in Lai et al. (2016) uses the centered second moment matrix to identify a sizable "good" subspace (spanned by the bottom eigenvectors) where the empirical mean achieves $O(\alpha)$ accuracy, and then it iterates on the remaining small subspace. Even in our more relaxed outlier model, the accuracy within the good subspace might be as bad as $\Omega(\alpha)$, which is insufficient for our purposes since we aim for $\epsilon$-error. Thus our Theorem 1.2 cannot be obtained by the technique of Lai et al. (2016) and we instead we have to look at the non-centered moment matrix after appropriate reweighting, as described in Section 1.2.

We conclude this section with two additional points of comparison.

**Connection to Mixture Models**    The mean-shift model is related to the classical task of parameter learning for mixture models, albeit in a regime that is qualitatively different from the one commonly studied. In the canonical setting (see Dasgupta (1999); Arora & Kannan (2001); Achlioptas & McSherry (2005); Kannan et al. (2005) for classic references and (Belkin & Sinha, 2015; Moitra & Valiant, 2010; Charikar et al., 2017; Hopkins & Li, 2018; Kothari et al., 2018; Diakonikolas et al., 2018b; Kong et al., 2020; Diakonikolas et al., 2022; Bakshi et al., 2022; Liu & Li, 2022; Diakonikolas et al., 2023) for more recent work), it is typically assumed that there is a small (constant) number of components, $k \ll n$, each with a distinct mean. In contrast, in the mean-shift model, all inlier samples come from the same component, while each outlier is drawn from its own component. Consequently, parameter estimation of the outlier means is information-theoretically impossible in the mean-shift model.

**Connection to Entangled Mean Estimation**    Another related contamination model in the context of mean estimation is the heteroskedasticity model. In heteroskedastic mean estimation, each datapoint is drawn independently from a potentially different distribution within a (known) family that shares a *common* mean. Such distributions are also referred to as *entangled*. For the Gaussian family, this model involves each sample having potentially different covariance (Chierichetti et al., 2014; Pensia et al., 2019a;b; 2021; Xia, 2019; Yuan & Liang, 2020; Liang & Yuan, 2020; Devroye et al., 2023; Compton & Valiant, 2024; Diakonikolas et al., 2025). This contamination model can also be viewed as a Gaussian mixture model. However, here each sample originates from its own component ($k = N$). Importantly, the shared mean assumption enables meaningful results despite the large number of components.

# B. Additional Preliminaries

**Cover set of the unit sphere**    For some of our proofs, we will need the following standard facts about cover sets of the unit sphere:

**Fact B.1** (see, e.g., Corollary 4.2.13 in Vershynin (2018))**.** *Let $\xi > 0$. There exists a set $\mathcal{C}$ of unit vectors of $\mathbb{R}^d$ such that $|\mathcal{C}| < (1 + 2/\xi)^d$ and for every $u \in \mathbb{R}^d$ with $\|u\| = 1$ it holds $\min_{y \in \mathcal{C}} \|y - u\| \le \xi$.*

**Corollary B.2** (see, e.g., Exercise 4.4.3 (b) in Vershynin (2018))**.** *There exists a subset $\mathcal{C}$ of the $d$-dimensional unit ball with $|\mathcal{C}| \le 7^d$ such that $\|x\| \le 2 \max_{v \in \mathcal{C}} |v^\top x|$ for all $x \in \mathbb{R}^d$ and $\|A\|_{\mathrm{op}} \le 3 \max_{x \in \mathcal{C}} x^\top A x$ for every symmetric $A \in \mathbb{R}^{d \times d}$.*

We will use the notions of subgaussian and subexponential random variables in the following standard way that we briefly review below.

**Subgaussian and subexponential random variables**    The following three statements are equivalent: (i) the random variable $x - \mathbf{E}[x]$ is subgaussian with "variance proxy" $\sigma^2$ (ii) $\|x - \mathbf{E}[x]\|_{L_p} \lesssim \sigma\sqrt{p}$ for all $p \ge 1$ and (iii) $\mathbf{Pr}[|x - \mathbf{E}[x]| > t] \le e^{-\Omega(t^2/\sigma^2)}$. The following three are also equivalent: (i) the random variable $x - \mathbf{E}[x]$ is subexponential with parameter $\lambda$ (ii) $\|x - \mathbf{E}[x]\|_{L_p} \lesssim p\lambda$ for all $p \ge 1$ and (iii) $\mathbf{Pr}[|x - \mathbf{E}[x]| > t] \le e^{-\Omega(t/\lambda)}$. If $X_1, \ldots, X_n$ are zero-mean independent subexponential random variables with parameter $\lambda$, and $\bar{X} = \sum_{i \in [n]} X_i/n$ then $\mathbf{Pr}[|\bar{X}| > t] \le \exp(-\Omega(n) \min(t^2/\lambda^2, t/\lambda))$ (Bernstein's inequality).

**Fact B.3** (see, e.g., Exercise 2.5.1 in Vershynin (2018))**.** *If $X \sim N(0, 1)$ then for any $p \ge 1$, $\|X\|_{L_p} = (\mathbf{E}[|X|^p])^{\frac{1}{p}} = \sqrt{2} \left[ \frac{\Gamma((1+p)/2)}{\Gamma(1/2)} \right]^{1/p}$, and thus by Stirling's approximation $\|X\|_{L_p} \lesssim \sqrt{p/e}$.*

**Fact B.4** (Matrix Bernstein Inequality; see, e.g., Theorem 5.4.1 in Vershynin (2018) )**.** *Let $X_1, \ldots, X_N$ be independent, mean zero, $n \times n$ symmetric random matrices, such that $\|X_i\|_{\mathrm{op}} \leq K$ almost surely for all $i$. Then, for every $t \geq 0$, we have*

$$\mathbb{P}\left[\left\|\sum_{i=1}^N X_i\right\|_{\mathrm{op}} \geq t\right] \leq 2n \exp\left(-c \cdot \min\left(\frac{t^2}{\sigma^2}, \frac{t}{K}\right)\right),$$

*where $\sigma^2 = \left\|\sum_{i=1}^N \mathbb{E}[X_i^2]\right\|_{\mathrm{op}}$ is the operator norm of the matrix variance of the sum.*

The following is a standard fact regarding concentration of norms for Gaussian vectors (see, e.g., Vershynin (2018) for a version of the fact for zero-mean Gaussians). For completeness, we provide a proof below.

**Fact B.5** (Gaussian Norm Concentration)**.** *If $x \sim \mathcal{N}(\mu, I)$, with probability $1 - \tau$ we have that*

$$\left|\|x\|^2 - \left(\|\mu\|^2 + d\right)\right| \lesssim \log\frac{1}{\tau} + \left(\sqrt{d} + \|\mu\|\right)\sqrt{\log\frac{1}{\tau}} \;.$$

*Proof.* Write $x = z + \mu$ for $z \sim \mathcal{N}(0, I)$ then

$$\left|\|z + \mu\|^2 - \|\mu\|^2 - d\right| = \left|\|z\|^2 - d + 2z^\top \mu\right| \leq \left|\|z\|^2 - d\right| + 2\left|z^\top \mu\right| \;.$$

First, using Bernstein's inequality, we have that

$$\left|\|z\|^2 - d\right| \lesssim \log\frac{1}{\tau} + \sqrt{d\log\frac{1}{\tau}}.$$

with probability at least $1 - \tau$. Also as $z^\top \mu \sim \mathcal{N}\left(0, \|\mu\|^2\right)$ it satisfies the Gaussian tails

$$\mathbf{Pr}\left[\left|z^\top \mu\right| > t\right] \leq e^{-\Omega\left(\frac{t^2}{\|\mu\|^2}\right)} \;,$$

or equivalently $\left|z^\top \mu\right| \lesssim \|\mu\|\sqrt{\log\frac{1}{\tau}}$ with probability $1 - \tau$. $\qquad\square$

## C. Omitted Proofs from Section 2.1

We restate and prove the following result.

**Proposition C.1** (Inefficient Estimator)**.** *Let $d \in \mathbb{Z}_+$ denote the dimension, and $C$ be a sufficiently large absolute constant. Let $\alpha \leq 0.49$, $\epsilon > 0$, and $\delta \in (0, 1)$ be parameters, and $\mu \in \mathbb{R}^d$ be an (unknown) vector. There exists an algorithm that, on input $\epsilon$ and any set of $n \geq 2^{C/\epsilon^2}(d + \log(1/\delta))$ $\alpha$-corrupted set of points from $\mathcal{N}(\mu, I)$ under the mean-shift model (cf. Definition 1.1), outputs a $\widehat{\mu}$ such that $\|\widehat{\mu} - \mu\| \leq \epsilon$ with probability at least $1 - \delta$. Moreover, it runs in time $2^{O(d)}\mathrm{poly}(n, d)$.*

*Proof of Proposition 2.1.* Denote by $T = \{x_i\}_{i=1}^n, x_i \in \mathbb{R}^d$ an $\alpha$-corrupted set of points from $\mathcal{N}(\mu, I)$ under the model of Definition 1.1 and denote by $C$ the the cover set of Corollary B.2. The algorithm is the following: First, using the algorithm from Fact 2.2, calculate a $m_v$ for each $v \in C$ such that $|m_v - v^\top \mu| \leq \varepsilon/8$ (see next paragraph for more details on this step). Then, output the solution of the following linear program (note that the program always has a solution, as it is satisfied by $\widehat{\mu} = \mu$):

$$\text{Find } \widehat{\mu} \in \mathbb{R}^d \text{ s.t.}$$
$$|v \cdot \widehat{\mu} - m_v| \leq \varepsilon/4, \forall v \in C \;.$$

The claim is that this solution $\widehat{\mu}$ is indeed close to the target $\mu$, since

$$\|\mu - \widehat{\mu}\| \leq 2\max_{v \in C}|v^\top(\mu - \widehat{\mu})| \qquad\qquad \text{(using Corollary B.2)}$$

$$\leq 2\max_{v \in C}(|v^\top \mu - m_v| + |m_v - v^\top\widehat{\mu}|)$$

$$\le 2(\epsilon/8 + \epsilon/4) < \epsilon . \tag{3}$$

We now explain how to obtain the approximations $m_v$ with the guarantee $|m_v - v^\top \mu| \le \varepsilon/8$. Fixing a direction $v \in \mathcal{C}$, we note that $v^\top x \sim \mathcal{N}(v^\top \mu, 1)$ thus $\{v^\top x_i\}_{i=1}^m$ is a set of $\alpha$-corrupted samples of $\mathcal{N}(v^\top \mu, 1)$. Thus, if we apply algorithm from Fact 2.2 with probability of failure $\delta' = \delta/|\mathcal{C}|$, the event $|m_v - v^\top \mu| \le \varepsilon/8$ will hold with probability at least $1 - \delta/|\mathcal{C}|$. By union bound, the probability all the events for $v \in \mathcal{C}$ hold simultaneously is at least $1 - \delta$. The number of samples for this application of Fact 2.2 is $2^{O(1/\epsilon^2)} \log(1/\delta') = 2^{O(1/\epsilon^2)} \log(|\mathcal{C}|/\delta) = 2^{O(1/\epsilon^2)}(d + \log(1/\delta))$.

We conclude with the runtime analysis. The runtime to find the $m_v$'s is $O(|\mathcal{C}| \operatorname{poly}(nd)) = 2^{O(d)} \operatorname{poly}(nd)$ since for each fixed $v \in \mathcal{C}$ we need $\operatorname{poly}(nd)$ time to calculate the projection $\{x_i^\top v\}$ of our dataset onto $v$ and $\operatorname{poly}(n)$ time to run the one-dimensional estimator. The linear program can be solved using the ellipsoid algorithm. Consider the separation oracle that exhaustively checks all $2^{O(d)}$ constraints. We need $\operatorname{poly}(d) \log(\frac{R}{r})$ calls to that separation oracle, where $R, r$ are the radii of the bounding spheres of the feasible region. First, $R \le \epsilon$, because we have already shown in (3) that the feasible set belongs in a ball of radius $\epsilon$ around $\mu$. Regarding the upper bound $r$, note that all $\widehat{\mu}$ inside a ball of radius $\varepsilon/8$ around $\mu$ are feasible since $|v^\top \widehat{\mu} - m_v| \le |v^\top \widehat{\mu} - v^\top \mu| + |v^\top \mu - m_v| \le \|\widehat{\mu} - \mu\| + \varepsilon/8 \le \epsilon/4$. This means that $r = \epsilon/4$. Hence the total runtime for solving the LP is $2^{O(d)} \operatorname{poly}(d)$ or simply $2^{O(d)}$.

$\square$

# D. Omitted Proofs from Section 2.2

In this section, we restate and prove Lemmas 2.3, 2.7 and 2.8.

**Lemma D.1.** *Let $\mathcal{V} \subseteq \mathbb{R}^d$ be a subspace of $\mathbb{R}^d$ of dimension $k \le d$. For any $z \in \mathcal{V}$, we have that $\mathbf{E}_{x \sim \mathcal{N}(z, \Pi_{\mathcal{V}})} [F_{\beta,k}(x)] Z_{\beta,k} = zz^\top e^{-\frac{\|z\|^2}{\beta\sqrt{k}+2}}$, where $F_{\beta,k}$ and $Z_{\beta,k}$ are as defined in Equation (1).*

*Proof.* We prove the lemma for $\mathcal{V} = \mathbb{R}^k$ which has $\Pi_{\mathcal{V}} = I$. The same proof generalizes to arbitrary subspaces. First, for any function $f : \mathbb{R}^k \to \mathbb{R}^{k \times k}$, we have that

$$
\begin{aligned}
\mathop{\mathbf{E}}_{x \sim \mathcal{N}(z, I)} \left[ f(x) e^{-\frac{\|x\|^2}{\beta\sqrt{k}}} \right] &= \frac{1}{Z} \int_{\mathbb{R}^k} f(x) e^{-\frac{\|x\|^2}{\beta\sqrt{k}}} e^{-\frac{\|x-z\|^2}{2}} dx && \text{(where } Z := (2\pi)^{k/2}) \\
&= \frac{1}{Z} \int_{\mathbb{R}^k} f(x) e^{-\frac{\|x\|^2}{\beta\sqrt{k}} - \frac{\|x-z\|^2}{2}} dx \\
&= \frac{1}{Z} \int_{\mathbb{R}^k} f(x) e^{-\frac{1}{2}(\|x\|^2(\frac{2}{\beta\sqrt{k}}+1) - 2z^\top x + \|z\|^2)} dx \\
&= \frac{1}{Z} \int_{\mathbb{R}^k} f(x) e^{-\frac{1}{2}(\|x\|^2 c - 2z^\top x + \|z\|^2)} dx && \text{(where } c := \frac{2}{\beta\sqrt{k}} + 1) \\
&= \frac{1}{Z} \int_{\mathbb{R}^k} f(x) e^{-\frac{c}{2}(\|x\|^2 - \frac{2}{c}z^\top x + \frac{1}{c}\|z\|^2)} dx \\
&= \frac{1}{Z} \int_{\mathbb{R}^k} f(x) e^{-\frac{c}{2}\left(\|x - \frac{1}{c}z\|^2 + (\frac{1}{c} - \frac{1}{c^2})\|z\|^2\right)} dx \\
&= \frac{e^{-\frac{1}{2}(1 - \frac{1}{c})\|z\|^2}}{Z} \int_{\mathbb{R}^k} f(x) e^{-\frac{c}{2}\|x - \frac{1}{c}z\|^2} dx \\
&= \frac{e^{-\frac{\|z\|^2}{\beta\sqrt{k}+2}}}{Z} \int_{\mathbb{R}^k} f(x) e^{-\frac{c}{2}\|x - \frac{1}{c}z\|^2} dx \\
&= \frac{e^{-\frac{\|z\|^2}{\beta\sqrt{k}+2}} Z'}{Z} \mathop{\mathbf{E}}_{x \sim \mathcal{N}(\frac{1}{c}z, \frac{1}{c}I)} [f(x)] && \text{(where } Z' := (2\pi)^{k/2}/c^{k/2}) \\
&= b \mathop{\mathbf{E}}_{x \sim \mathcal{N}(\frac{1}{c}z, \frac{1}{c}I)} [f(x)] . && \text{(where } b := e^{-\frac{\|z\|^2}{\beta\sqrt{k}+2}}(1/c)^{k/2})
\end{aligned}
$$

Therefore, for $f(x) = xx^\top - (1/c)I$, we have that

$$
\mathop{\mathbf{E}}_{x \sim \mathcal{N}(z,I)} \left[ \left( xx^\top - \frac{1}{c}I \right) e^{-\frac{\|x\|^2}{\beta\sqrt{k}}} \right] = \mathop{\mathbf{E}}_{x \sim \mathcal{N}(z,I)} \left[ f(x) e^{-\frac{\|x\|^2}{\beta\sqrt{k}}} \right] = b \mathop{\mathbf{E}}_{x \sim \mathcal{N}(\frac{1}{c}z, \frac{1}{c}I)} [f(x)]
$$

$$
= b \left( \mathop{\mathbf{E}}_{x \sim \mathcal{N}(\frac{1}{c}z, \frac{1}{c}I)} [xx^\top] - \frac{1}{c}I \right) = b \left( \frac{1}{c}I + \frac{1}{c^2} zz^\top - \frac{1}{c}I \right)
$$

$$
= b \left( \frac{1}{c^2} zz^\top \right) = e^{-\frac{\|z\|^2}{\beta\sqrt{k}+2}} \left( \frac{1}{1 + \frac{2}{\beta\sqrt{k}}} \right)^{\frac{k}{2}+2} zz^\top .
$$

$\square$

**Lemma D.2.** *Let $\eta \in (0,1)$ and $k \in \mathbb{Z}_+$ denote the dimension, and assume $k$ is bigger than a sufficiently large constant. There exists sample size $n = k\log^3(k)(1/\eta)^{2+o(1)}\frac{1}{\delta}$ such that the following holds: Let $T$ be a set of $n$ $\alpha$-corrupted points from $\mathcal{N}(\mu, I)$ according to Definition 1.1 with the assumption that $\|\mu\| = O(1)$. Denote by $z_1, \ldots, z_{\alpha n}$ the adversarial centers in Definition 1.1. Then, with probability at least $1 - \delta$, $T$ is $(\eta, \sqrt{\log k})$-good with respect to $\mu, z_1, \ldots, z_{\alpha n}$.*

*Proof.* We will show that Item 2 and Item 1 of Definition 2.6 hold. Without loss of generality, we use $\mathcal{V} = \mathbb{R}^k$ in Definition 2.6, i.e., our subspace is the entire $\mathbb{R}^k$. We start with Item 2 since it is more general. The other will be similar.

**Proof of Item 2** We remind the reader that $T = \{x_i, \ldots, x_{|T|}\}$ is the set are the $\alpha$-corrupted points and that $T \setminus S$ denotes the subset of $\alpha n$ outliers with $z_i$ the associated center of $x_i \in T \setminus S$. We want to show that the sample average over $T \setminus S$ matrix $\widehat{A}$ from (1), with $\beta = \sqrt{\log(k)}$, satisfies $v^\top \widehat{A} v > -\eta$ for all unit vectors $v$. Without loss of generality, instead of the definition of $\widehat{A}$ given in (1) we will use the following definition:

$$
\widehat{A} = \frac{1}{|T \setminus S|} \sum_{i: x_i \in T \setminus S} \widehat{A}_i \quad \text{where} \quad \widehat{A}_i = F_{\beta,k}(x_i) e^{\sqrt{k/\log(k)}} \mathbb{1}(\mathcal{E}_i) , \tag{4}
$$

where $F_{\beta,k}$ as in Equation (1) and $\mathcal{E}_i$ is defined to be the following *good event*:

$$
\left| \|x_i\|^2 - \left( \|z_i\|^2 + k \right) \right| \lesssim \log\frac{1}{\tau} + \left( \sqrt{k} + \|z_i\| \right) \sqrt{\log\frac{1}{\tau}} , \tag{5}
$$

where $\tau := (1/n)^4$. The first change in our definition is that we replaced the normalization factor $Z_{\beta,k}$ that appeared in (1) by the simpler expression $e^{\sqrt{k/\log(k)}}$. This is because $(1 + 2/(\sqrt{k\log(k)}))^{k/2+2} = \Theta(e^{\sqrt{k/\log(k)}})$, and using $e^{\sqrt{k/\log(k)}}$ will be more convenient for our calculations later on. Without loss of generality we can use $\Pi_\mathcal{V} = I_k$ as we can all of the statements that we aim to prove are similarity invariant. The second change is that we are using the good event in the definition of $\widehat{A}$. This is indeed without loss of generality because (i) by Fact B.5 we know that $\mathcal{E}_i$ holds for each sample $x_i$ with probability $1 - \tau$, thus by a union bound over all samples we have that all the events $\mathcal{E}_i$ hold simultaneously with probability at least $1 - n\tau \geq 1 - 1/n^3 > 1 - \delta/8$ (for $k \geq 2$), and (ii) using the indicator $\mathbb{1}(\mathcal{E}_i)$ in the definition of $\widehat{A}$ shifts the expected value of $\widehat{A}$ by a negligible amount (much smaller than $\eta$) as we show below:

$$
\left| v^\top \mathop{\mathbf{E}}_{x_i \sim \mathcal{N}(z_i, I)} \left[ F_{\beta,k}(x_i) e^{\sqrt{k/\log(k)}} \mathbb{1}(\mathcal{E}_i) - F_{\beta,k}(x_i) e^{\sqrt{k/\log(k)}} \right] v \right|
$$

$$
= \left| v^\top \mathop{\mathbf{E}}_{x_i \sim \mathcal{N}(z_i, I)} \left[ \left( x_i x_i^\top - \frac{\sqrt{k\log(k)}}{\sqrt{k\log(k)}+2} I \right) e^{-\frac{\|x_i\|^2-k}{\sqrt{k\log(k)}}} \mathbb{1}(\bar{\mathcal{E}}_i) \right] v \right|
$$

$$
\leq \mathop{\mathbf{E}}_{x_i \sim \mathcal{N}(z_i, I)} \left[ \left| (v^\top x_i)^2 - \frac{\sqrt{k\log(k)}}{\sqrt{k\log(k)}+2} \right| e^{-\frac{\|x_i\|^2-k}{\sqrt{k\log(k)}}} \mathbb{1}(\bar{\mathcal{E}}_i) \right]
$$

$$
\leq \sqrt{\mathop{\mathbf{Pr}}_{x_i \sim \mathcal{N}(z_i, I)}[\bar{\mathcal{E}}_i]} \sqrt{\mathop{\mathbf{E}}_{x_i \sim \mathcal{N}(z_i, I)} \left[ \left( \left( (v^\top x_i)^2 - \frac{\sqrt{k\log(k)}}{\sqrt{k\log(k)}+2} \right) e^{-\frac{\|x_i\|^2-k}{\sqrt{k\log(k)}}} \right)^2 \right]} \quad \text{(using Cauchy-Schwarz)}
$$

$$\leq \left(\frac{1}{n}\right)^2 \sqrt{\underset{x_i \sim \mathcal{N}(z_i, I)}{\mathbf{E}}\left[\left(\left((v^\top x_i)^2 - \frac{\sqrt{k\log(k)}}{\sqrt{k\log(k)}+2}\right)e^{-\frac{\|x_i\|^2-k}{\sqrt{k\log(k)}}}\right)^2\right]} \qquad \text{(using Fact B.5 with } \tau = (1/n)^4)$$

$$\lesssim \left(\frac{1}{n}\right)^2 \sqrt{k}\log(k)e^{2/\log^2(k)} . \qquad\qquad\qquad \text{(using Claim D.6 applied with } \beta = \sqrt{\log(k)})$$

$$\leq \eta/2 , \tag{6}$$

where Claim D.6 that was used above is a bound on the reweighted second moment that can be found in Appendix D.1. The above is considering only a single term in the definition of $\widehat{A}$. By triangle inequality it also follows that $\left\|\mathbf{E}\left[\widehat{A}\right] - A\right\| \leq \eta/2$, where $A = \mathbf{E}\left[\frac{1}{|T\setminus S|}\sum_{i:x_i \in T\setminus S} F_{\beta,k}(x_i)e^{\sqrt{k/\log(k)}}\mathbb{1}(\mathcal{E}_i)\right]$.

We now move to show that $v^\top \widehat{A} v > -\eta$ with high probability. Write $x_i = z_i + g_i$ where $g_i \sim \mathcal{N}(0, I)$ for $i \in T \setminus S$. We decompose the matrix $\widehat{A}_i$ for $i : x_i \in T \setminus S$ into three terms

$$\widehat{A}_i = \left(\left(g_i g_i^\top - \frac{\sqrt{k\log(k)}}{\sqrt{k\log(k)}+2}I\right) + (z_i g_i^\top + g_i z_i^\top) + z_i z_i^\top\right)e^{-\frac{\|x_i\|^2-k}{\sqrt{k\log(k)}}}\mathbb{1}\left(\mathcal{E}_i\right)$$

$$=: \widehat{T}_{1i} + \widehat{T}_{2i} + \widehat{T}_{3i} ,$$

where by $\widehat{T}_{1i}, \widehat{T}_{2i}$ and $\widehat{T}_{3i}$ we denote each of the terms that sum to $\widehat{A}_i$ and by $T_{1i}, T_{2i}$ and $T_{3i}$ we denote their corresponding expectations. We will show concentration for each of the terms separately and then combine the results to show that $v^\top \widehat{A} v > -\eta$.

**First Term** Fix $v \in \mathbb{R}^k : \|v\| = 1$ and let $i$ such that $x_i \in T \setminus S$. We bound the $L_p$ norm of the random variable $v^\top\left(\widehat{T}_{1i} - T_{1i}\right)v$ as follows:

$$\left\|v^\top\left(\widehat{T}_{1i} - T_{1i}\right)v\right\|_{L_p} \lesssim \left\|v^\top \widehat{T}_{1i}v\right\|_{L_p} \qquad\qquad \text{(by triangle inequality and Jensen's inequality)}$$

$$= \left\|\left((g_i^\top v)^2 - \frac{\sqrt{k\log(k)}}{\sqrt{k\log(k)}+2}\right)e^{-\frac{\|x_i\|^2-k}{\sqrt{k\log(k)}}}\mathbb{1}(\mathcal{E}_i)\right\|_{L_p}$$

$$\leq \left\|\left((g_i^\top v)^2 - \frac{\sqrt{k\log(k)}}{\sqrt{k\log(k)}+2}\right)\mathbb{1}\left(\mathcal{E}_i\right)\right\|_{L_p}(1/\eta)^{o(1)}e^{-\frac{\|z_i\|^2}{\sqrt{k\log(k)}}+O\left(\frac{\|z_i\|\sqrt{\log(k/\eta)}}{\sqrt{k\log(k)}}\right)} \qquad \text{(by Claim D.3)}$$

$$\leq \left(\left\|(g_i^\top v)^2\right\|_{L_p} + \frac{\sqrt{k\log(k)}}{\sqrt{k\log(k)}+2}\right)(1/\eta)^{o(1)}e^{O\left(\frac{\log(k/\eta)}{\sqrt{k\log(k)}}\right)} \qquad \text{(by Claim D.4)}$$

$$\lesssim \left(p + \frac{\sqrt{k\log(k)}}{\sqrt{k\log(k)}+2}\right)(1/\eta)^{o(1)} \qquad\qquad \text{(by Fact B.3 since } g_i^\top v \sim N(0,1))$$

$$\lesssim p \cdot (1/\eta)^{o(1)} , \tag{7}$$

where the third line above uses the definition of the good event $\mathcal{E}_i$. The proof follows by the definition of the events $\mathcal{E}_i$ and some simple algebra. We include the proofs of both claims in Appendix D.1.

**Claim D.3.** *Let $\mathcal{E}_i$ denote the event from* (5)*. Then,*

$$\exp\left(-\frac{\|x_i\|^2-k}{\sqrt{k\log(k)}}\right)\mathbb{1}(\mathcal{E}_i) \leq \exp\left(-\frac{\|z_i\|^2}{\sqrt{k\log(k)}} + O\left(\frac{\|z_i\|\sqrt{\log(k/\eta)}}{\sqrt{k\log(k)}}\right)\right)(1/\eta)^{o(1)} .$$

**Claim D.4.** *Fix a $k > 10, 0 < \eta < 1$ and $C > 8$. The following inequalities hold:*

*i.* $-\frac{x^2}{\sqrt{k\log(k)}} + C\frac{x\sqrt{\log\left(\frac{k}{\eta}\right)}}{\sqrt{k\log(k)}} \leq C^2 \frac{\log\left(\frac{k}{\eta}\right)}{\sqrt{k\log(k)}}$ *for all $x > 0$.*

*ii.* $x^2 e^{-\frac{x^2}{\sqrt{k\log(k)}} + C\frac{x\sqrt{\log\frac{k}{\eta}}}{\sqrt{k\log(k)}}} \lesssim C^2 \sqrt{k\log k}\log\left(\frac{k}{\eta}\right)e^{C^2\frac{\log\left(\frac{k}{\eta}\right)}{\sqrt{k\log(k)}}}$ *for all $x > 0$.*

*iii.* $x^4 e^{-\frac{2x^2}{\sqrt{k\log(k)}} + C\frac{x\sqrt{\log(k/\eta)}}{\sqrt{k\log(k)}}} \lesssim C^4 k\log(k)\log^2\left(\frac{k}{\eta}\right)e^{C^2\frac{\log\left(\frac{k}{\eta}\right)}{\sqrt{k\log(k)}}}$ *for all $x > 0$.*

As a result of (7), we have that $v^\top\left(\widehat{T}_{1i} - T_{1i}\right)v$ (where $T_{1i}$ denotes the expectation of $\widehat{T}_{1i}$) is sub-exponential random variable with parameter $\lambda = \frac{1}{\eta^{o(1)}}$, hence from Bernstein's inequality we have that

$$\mathbf{Pr}\left[\left|\frac{1}{\alpha n}\sum_{i:x_i \in T\setminus S}v^\top\left(\widehat{T}_{1i} - T_{1i}\right)v\right| \geq t\right] \leq 2\exp\left(-c\min\left(\frac{t^2\alpha n}{\lambda^2}, \frac{t\alpha n}{\lambda}\right)\right),$$

Using the above with $t = \eta/6$, we have that with $n = \frac{\log(1/\delta')}{\eta^{2+o(1)}}$ samples[3] we have that with probability at least $1 - \delta'$ for a fixed vector $v$ it holds that $\left|\frac{1}{\alpha n}\sum_{i:x_i \in T\setminus S}v^\top\left(\widehat{T}_{1i} - T_{1i}\right)v\right| \leq O(\eta)$. Now let $\mathcal{C}$ be a cover of the unit ball from Corollary B.2. By using $\delta' = \frac{\delta}{8|\mathcal{C}|} = \delta 2^{-O(k)}$ and a union bound over $\mathcal{C}$ we have that

$$\sup_{v\in\mathbb{R}^k:\|v\|=1}\left|\frac{1}{\alpha n}\sum_{i:x_i \in T\setminus S}v^\top\left(\widehat{T}_{1i} - T_{1i}\right)v\right| \leq 10\max_{v\in\mathcal{C}}\left|\frac{1}{\alpha n}\sum_{i:x_i \in T\setminus S}v^\top\left(\widehat{T}_{1i} - T_{1i}\right)v\right|,$$

which can be made less than $\eta/6$ with probability $1 - \delta/8$ by using $n = \frac{d\log(1/\delta)}{\eta^{2+o(1)}}$ samples.

**Third Term** For the third term $\widehat{T}_{3i} = z_i z_i^\top e^{-\frac{\|x_i\|^2 - k}{\sqrt{k\log(k)}}}$, we will use the Matrix Bernstein Inequality (Fact B.4). Recall our notation that $T_{3i}$ denotes the expected value of the random matrix $\widehat{T}_{3i}$. First, we have that

$$\left\|\widehat{T}_{3i} - T_{3i}\right\|_{op} \lesssim \left\|\widehat{T}_{3i}\right\|_{op} = \left\|z_i z_i^\top e^{-\frac{\|x_i\|^2 - k}{\sqrt{k\log(k)}}}\mathbb{1}\left(\mathcal{E}_i\right)\right\|_{op}$$

$$= \left\|z_i z_i^\top\right\|_{op} e^{-\frac{\|x_i\|^2 - k}{\sqrt{k\log(k)}}}\mathbb{1}\left(\mathcal{E}_i\right)$$

$$= \|z_i\|^2 e^{-\frac{\|z_i\|^2}{\sqrt{k\log(k)}} + O\left(\frac{\|z_i\|\sqrt{\log(k/\eta)}}{\sqrt{k\log(k)}}\right)}(1/\eta)^{o(1)} \qquad \text{(using Claim D.3)}$$

$$\lesssim \sqrt{k\log(k)}\log(k/\eta)(1/\eta)^{o(1)}, \qquad \text{(by Claim D.4)}$$

$$\lesssim \sqrt{k}\log^{3/2}(k)\log(1/\eta)(1/\eta)^{o(1)}$$

$$\lesssim \sqrt{k}\log^{3/2}(k)(1/\eta)^{o(1)},$$

where the first inequality follows by the triangle inequality and Jensen's inequality almost surely. Moreover

$$\left\|\mathbf{E}[\left(\widehat{T}_{3i} - T_{3i}\right)^2]\right\|_{op} \leq \left\|\mathbf{E}[\widehat{T}_{3i}^2]\right\|$$

$$= \left\|\mathbf{E}\left[\|z_i\|^2 z_i z_i^\top e^{-2\frac{\|x_i\|^2 - k}{\sqrt{k\log(k)}}}\mathbb{1}\left(\mathcal{E}_i\right)\right]\right\|_{op}$$

---

[3]Here we have used that without loss of generality $\alpha = \Omega(1)$ since we can always treat some of the inliers as outliers in the model of Definition 1.1

$$= \|z_i\|^2 \left\| z_i z_i^\top \right\|_{\mathrm{op}} \mathbf{E}\left[ e^{-2\frac{\|x_i\|^2 - k}{\sqrt{k\log(k)}}} \mathbb{1}\left( \mathcal{E}_i \right) \right]$$

$$\leq \|z_i\|^4 e^{-\frac{2\|z_i\|^2}{\sqrt{k\log(k)}} + O\left( \frac{\|z_i\|\sqrt{\log(k/\eta)}}{\sqrt{k\log(k)}} \right)} (1/\eta)^{o(1)} \qquad \text{(using Claim D.3)}$$

$$\leq k\log(k)\log^2(k/\eta) e^{O\left( \frac{\log(k/\eta)}{\sqrt{k\log(k)}} \right)} (1/\eta)^{o(1)}. \qquad \text{(by Claim D.4)}$$

$$\leq k\log^3(k)\log^2(1/\eta) e^{O\left( \frac{\log(1/\eta)}{\sqrt{k\log(k)}} \right)} (1/\eta)^{o(1)}$$

$$\leq k\log^3(k)(1/\eta)^{o(1)} .$$

Hence, from Fact B.4, we have that

$$\mathbf{Pr}\left[ \left\| \frac{1}{\alpha n} \sum_{i:x_i \in T\setminus S} \left( \widehat{T}_{3i} - T_{3i} \right) \right\|_{\mathrm{op}} \geq t \right] \leq 2k \exp\left( -c \cdot \min\left( \frac{t^2 \alpha n}{k\log^3(k)(1/\eta)^{o(1)}}, \frac{t\alpha n}{\sqrt{k}\log^{3/2}(k)(1/\eta)^{o(1)}} \right) \right).$$

In summary, using $n = O\left( (1/\eta)^{2+o(1)} k\log^3(k)\log(1/\delta) \right)$, we have that $\left\| \frac{1}{\alpha n} \sum_{i:x_i \in T\setminus S} \left( \widehat{T}_{3i} - T_{3i} \right) \right\|_{\mathrm{op}} \leq \eta/6$ with probability at least $1 - \delta/8$.

**Second Term** For the term $\widehat{T}_2$ we will prove a multiplicative bound. Again, fix a direction $v$ with $\|v\| = 1$. We will first show that $v^\top \widehat{T}_2 v$ is subgaussian by bounding the $L_p$-norms:

$$\left\| v^\top \left( \widehat{T}_{2i} - T_{2i} \right) v \right\|_{L_p} \lesssim \left\| v^\top \widehat{T}_{2i} v \right\|_{L_p} = \left\| \left( g_i^\top v z_i^\top v \right) e^{-\frac{\|x_i\|^2 - k}{\sqrt{k\log(k)}}} \mathbb{1}(\mathcal{E}_i) \right\|_{L_p}$$

$$\leq \left\| g_i^\top v \right\|_{L_p} \left| v^\top z_i \right| e^{-\frac{\|z_i\|^2}{\sqrt{k\log(k)}} + O\left( \frac{\|z_i\|\sqrt{\log(k/\eta)}}{\sqrt{k\log(k)}} \right)} (1/\eta)^{o(1)} \quad \text{(using Claim D.3)}$$

$$\lesssim \sqrt{p} \left| v^\top z_i \right| e^{-\frac{\|z_i\|^2}{\sqrt{k\log(k)}} + O\left( \frac{\|z_i\|\sqrt{\log(k/\eta)}}{\sqrt{k\log(k)}} \right)} (1/\eta)^{o(1)} . \quad \text{(since } g_i^\top v \sim \mathcal{N}(0, I) \text{)}$$

Hence, it follows that $v^\top \left( \widehat{T}_{2i} - T_{2i} \right) v$ are independent subgaussian random variables with proxy standard deviations $\sigma_i = \left| v^\top z_i \right| e^{-\frac{\|z_i\|^2}{\sqrt{k\log(k)}} + O\left( \frac{\|z_i\|\sqrt{\log(k/\eta)}}{\sqrt{k\log(k)}} \right)} (1/\eta)^{o(1)}$. Thus, the proxy variance $\sigma^2$ of the average $(1/\alpha n) \sum_{i:x_i T\setminus S} v^\top \left( \widehat{T}_{2i} - T_{2i} \right) v$ is

$$\sigma^2 \lesssim \frac{1}{(\alpha n)^2} \sum_{i:x_i \in T\setminus S} (v^\top z_i)^2 e^{-2\frac{\|z_i\|^2}{\sqrt{k\log(k)}} + O\left( \frac{\|z_i\|\sqrt{\log(k/\eta)}}{\sqrt{k\log(k)}} \right)} (1/\eta)^{o(1)}$$

$$\lesssim \frac{1}{(\alpha n)^2} \sum_{i:x_i \in T\setminus S} (v^\top z_i)^2 e^{-\|z_i\|^2/\sqrt{k\log(k)}} (1/\eta)^{o(1)}$$

$$\leq \frac{1}{(\alpha n)^2} \sum_{i:x_i \in T\setminus S} (v^\top z_i)^2 e^{-\|z_i\|^2/(\sqrt{k\log(k)}+2)} (1/\eta)^{o(1)}$$

$$\lesssim \frac{1}{\alpha n} v^\top A v (1/\eta)^{o(1)} , \qquad \text{(for } A := \frac{1}{\alpha n} \sum_{i:x_i \in T\setminus S} z_i z_i^\top e^{-\frac{\|z_i\|^2}{\sqrt{k\log(k)}+2}} \text{)}$$

where the second line used the following:

$$-2\frac{\|z_i\|^2}{\sqrt{k\log(k)}} + O\left( \frac{\|z_i\|\sqrt{\log(k/\eta)}}{\sqrt{k\log(k)}} \right) = -\frac{\|z_i\|^2}{\sqrt{k\log(k)}} + \left( O\left( \frac{\|z_i\|\sqrt{\log(k/\eta)}}{\sqrt{k\log(k)}} \right) - \frac{\|z_i\|^2}{\sqrt{k\log(k)}} \right)$$

$$\leq -\frac{\|z_i\|^2}{\sqrt{k\log(k)}} + O\left((k\log(k))^{-1/2}\log(k/\eta)\right)$$

(using Claim D.4 for last two terms)

$$\leq -\frac{\|z_i\|^2}{\sqrt{k\log(k)}} + O\left(\frac{\log(1/\eta)}{\sqrt{k\log k}}\right) .$$

Using the subgaussian tails of the random variable $\frac{1}{\alpha n}\sum_i v^\top\left(\widehat{T}_{2i} - T_{2i}\right)v$, we have that with probability $1 - \delta'$ it holds

$$\left|\frac{1}{\alpha n}\sum_{i:x_i\in T\backslash S} v^\top\left(\widehat{T}_{2i} - T_{2i}\right)v\right| \lesssim \sigma\sqrt{\log(1/\delta')} \lesssim \sqrt{\frac{1}{\alpha n}(v^\top Av)\,(1/\eta)^{o(1)}\log(1/\delta')} . \tag{8}$$

Now let $\mathcal{C}$ be a cover of the unit ball with accuracy $\xi = \eta/k$. The size of such a cover is $|\mathcal{C}| = (O(k/\eta))^k$ (Fact B.1). If we use $\delta' = \delta/(8|\mathcal{C}|)$ and do a union bound over $\mathcal{C}$ we have that (8) holds with probability $1 - \delta/8$ for all $v \in \mathcal{C}$ simultaneously. By plugging in the aforementioned value for $\delta'$, we get that the following holds for all $v \in \mathcal{C}$:

$$\left|\frac{1}{\alpha n}\sum_{i:x_i\in T\backslash S} v^\top\left(\widehat{T}_{2i} - T_{2i}\right)v\right| \lesssim \sqrt{\frac{1}{\alpha n}(v^\top Av)\,(1/\eta)^{o(1)}k\log(k/\eta)\log((1/\delta)}$$

$$\lesssim \sqrt{\frac{1}{\alpha n}(v^\top Av)\,(1/\eta)^{o(1)}k\log(1/\delta)} .$$

By using $\alpha n = O\left(k(1/\eta)^{o(1)}\log(1/\delta)\right)$ samples, the above implies that for all $v \in \mathcal{C}$ it holds

$$\frac{1}{\alpha n}\sum_{i:x_i\in T\backslash S} v^\top\widehat{T}_{2i}v \geq \frac{1}{\alpha n}\sum_{i:x_i\in T\backslash S} v^\top T_{2i}v - (\eta/6)\sqrt{v^\top Av} .$$

As a result, combining the bounds for the three terms, we have that for every $v \in \mathcal{C}$:

$$v^\top\widehat{A}v = \frac{1}{\alpha n}\sum_{i:x_i\in T\backslash S} v^\top\left(\widehat{T}_{1i} + \widehat{T}_{2i} + \widehat{T}_{3i}\right)v$$

$$\geq \frac{1}{\alpha n}\sum_{i:x_i\in T\backslash S} v^\top\left(T_{1i} + T_{2i} + T_{3i}\right)v - \eta/3 - (\eta/6)\sqrt{v^\top Av}$$

$$= v^\top \mathbf{E}[\widehat{A}]v - \eta/3 - (\eta/6)\sqrt{v^\top Av}$$

$$\geq v^\top Av - \frac{5}{6}\eta - (\eta/6)\sqrt{v^\top Av} . \tag{by (6)}$$

From this, it now easily follows that $v^\top\widehat{A}v \geq -\eta$ by a simple case analysis: If $v^\top Av > 1$, then

$$v^\top Av - \frac{5}{6}\eta - (\eta/6)\sqrt{v^\top Av} \geq v^\top Av - \frac{5}{6}\eta - \sqrt{v^\top Av} \tag{$\eta < 1$}$$

$$\geq -\frac{5}{6}\eta ,$$

where the last line uses that $v^\top Av - \sqrt{v^\top Av} \geq 0$ whenever $v^\top Av > 1$. If on the other hand $v^\top Av \leq 1$, then our bound becomes $v^\top Av - \frac{5}{6}\eta - (\eta/6)\sqrt{v^\top Av} \geq v^\top Av - \frac{5}{6}\eta - (\eta/6) = v^\top Av - \eta \geq -\eta$, where we also used that $v^\top Av \geq 0$ by definition of $A := \frac{1}{\alpha n}\sum_{i:x_i\in T\backslash S} z_iz_i^\top e^{-\frac{\|z_i\|^2}{\sqrt{k\log(k)}+2}}$.

So far we have shown that $v^\top\widehat{A}v \geq -\eta$ for every $v \in \mathcal{C}$, where $\mathcal{C}$ is a cover of the $k$-dimensional unit ball with accuracy $\xi = \eta/k$. It is easy to see that this implies that $u^\top\widehat{A}u \geq -4\eta$ for all arbitrary $u$ with $\|u\| = 1$. Consider an arbitrary unit vector $u$. There exists a $v \in \mathcal{C}$ such that $\|v - u\| \leq \xi$ i.e., $u = v + w$, with $\|w\| \leq \xi$. Thus

$u^\top \widehat{A} u = v^\top \widehat{A} v + v^\top A w + w^\top A v + w^\top \widehat{A} w \geq -\eta - |v^\top A w| - |w^\top A v| - |w^\top \widehat{A} w|$. Since $\|A\|_{\text{op}} \leq \sqrt{k \log(k)}$ and $\|w\| \leq \xi = \eta/k$, $|u^\top A w| \leq \|w\| \|A\|_{\text{op}} \leq \eta$ and $|w^\top A w| \leq \xi^2 \eta \leq \eta$.

The total probability of failure for Item 2 is $\delta/2$ by the union bound as the proof relies on 4 events with probability of failure $\delta/8$.

**Proof of Item 1**  We will use a similar argument as in Item 2. Let $\mathcal{E}_i$ be the same as in (5). First, we will use

$$\widehat{A} = \frac{1}{|S|} \sum_{i:x_i \in S} \widehat{A}_i \quad \text{where} \quad \widehat{A}_i = F_{\beta,k}(x_i) e^{\sqrt{k/\log(k)}} \mathbb{1}(\mathcal{E}_i) \ .$$

The points $x_i \in S$ are all drawn from the same Gaussian component $\mathcal{N}(\mu, I)$. Fix $v \in \mathbb{R}^k : \|v\| = 1$. We bound the $L_p$ norm of the random variable $v^\top \left( \widehat{A}_i - \mathbf{E}[A_i] \right) v$ for $i : x_i \in S$ as follows:

$$\left\| v^\top \left( \widehat{A}_i - \mathbf{E}[A_i] \right) v \right\|_{L_p} \tag{9}$$

$$\lesssim \left\| v^\top \widehat{A}_i v \right\|_{L_p} \qquad \text{(by triangle inequality and Jensen's inequality)}$$

$$= \left\| \left( (x_i^\top v)^2 - \frac{\sqrt{k \log(k)}}{\sqrt{k \log(k)} + 2} \right) e^{-\frac{\|x_i\|^2 - k}{\sqrt{k \log(k)}}} \mathbb{1}(\mathcal{E}_i) \right\|_{L_p}$$

$$\leq \left\| (x_i^\top v)^2 - \frac{\sqrt{k \log(k)}}{\sqrt{k \log(k)} + 2} \right\|_{L_p} e^{-\frac{\|\mu\|^2}{\sqrt{k \log(k)}} + O\left( \frac{\|\mu\| \sqrt{\log(k/\eta)}}{\sqrt{k \log(k)}} \right)} (1/\eta)^{o(1)} \qquad \text{(by Claim D.3)}$$

$$\leq \left( \left\| (\mu^\top v + g_i)^2 \right\|_{L_p} + \frac{\sqrt{k \log(k)}}{\sqrt{k \log(k)} + 2} \right) e^{-\frac{\|\mu\|^2}{\sqrt{k \log(k)}} + O\left( \frac{\|\mu\| \sqrt{\log(k/\eta)}}{\sqrt{k \log(k)}} \right)} (1/\eta)^{o(1)} \qquad (x_i = g_i + \mu \text{ for } g_i \sim N(0,1))$$

$$\leq \left( (\mu^\top v)^2 + \left\| g_i^2 \right\|_{L_p} + \frac{\sqrt{k \log(k)}}{\sqrt{k \log(k)} + 2} \right) e^{-\frac{\|\mu\|^2}{\sqrt{k \log(k)}} + O\left( \frac{\|\mu\| \sqrt{\log(k/\eta)}}{\sqrt{k \log(k)}} \right)} (1/\eta)^{o(1)}$$

$$\lesssim \left( \|\mu\|^2 + p + \frac{\sqrt{k \log(k)}}{\sqrt{k \log(k)} + 2} \right) e^{-\frac{\|\mu\|^2}{\sqrt{k \log(k)}} + O\left( \frac{\|\mu\| \sqrt{\log(k/\eta)}}{\sqrt{k \log(k)}} \right)} (1/\eta)^{o(1)} \qquad \text{(by Fact B.3)}$$

$$\lesssim p \|\mu\|^2 e^{-\frac{\|\mu\|^2}{\sqrt{k \log(k)}} + O\left( \frac{\|\mu\| \sqrt{\log(k/\eta)}}{\sqrt{k \log(k)}} \right)} (1/\eta)^{o(1)}$$

$$\lesssim p (1/\eta)^{o(1)} \ . \qquad (\|\mu\| = O(1))$$

Hence, by a cover argument identical to the one that we used when treating the first term in Item 2, we have that with $n = O\left( \frac{k \log(1/\delta)}{\eta^{2+o(1)}} \right)$ we have that $\left\| \widehat{A} - \mathbf{E}[A] \right\|_{\text{op}} \leq \eta/2$ and hence $\left\| \widehat{A} - A \right\|_{\text{op}} \leq \eta$ with probability at least $1 - \delta/2$ where $A := \mu \mu^\top e^{-\frac{\|\mu\|^2}{\sqrt{k \log(k)} + 2}}$.

**Proof of Item 3**  Let $\widehat{A}$ the sample average matrix from (1) over the whole sample set $T$ with $\beta = 1$, we will show that $\text{tr}(\widehat{A}) \leq 18 \sqrt{k \log(k)}$. In fact, we will show that this holds with probability 1.

$$\text{tr}(\widehat{A}) = \frac{1}{n} \sum_{i \in [n]} \left( \|x_i\|^2 - \frac{\sqrt{k \log(k)}}{\sqrt{k \log(k)} + 2} k \right) e^{-\frac{\|x_i\|^2}{\sqrt{k \log(k)}}} (1 + 2/\sqrt{k \log(k)})^{k/2 + 2}$$

$$= \frac{1}{n} \sum_{i \in [n]} \left( \|x_i\|^2 - \frac{\sqrt{k \log(k)}}{\sqrt{k \log(k)} + 2} k \right) e^{-\frac{\|x_i\|^2 - k}{\sqrt{k \log(k)}}} e^{-\sqrt{k/\log(k)}} (1 + 2/\sqrt{k \log(k)})^{k/2 + 2} \ .$$

We will show that each term in that sum individually is at most $18\sqrt{k\log(k)}$. This will follow by two facts: (i) we have that $e^{-\sqrt{k/\log(k)}}(1 + 2/\sqrt{k\log(k)})^{k/2+2} \leq 6$ for all $k \geq 10$, and (ii) it is true that $(\|x_i\|^2 - k\sqrt{k\log(k)}/(2 + \sqrt{k\log(k)}))e^{-\frac{\|x_i\|^2 - k}{\sqrt{k\log(k)}}} \leq 3\sqrt{k\log(k)}$, which we will show in what follows. Let $y := \|x_i\|^2 - \frac{\sqrt{k\log(k)}}{\sqrt{k\log(k)}+2}k$. We have that

$$\left(\|x_i\|^2 - \frac{\sqrt{k\log(k)}}{\sqrt{k\log(k)} + 2}k\right)e^{-\frac{\|x_i\|^2 - k}{\sqrt{k\log(k)}}} = ye^{-\frac{y}{\sqrt{k\log(k)}}}e^{\frac{2\sqrt{k}}{\sqrt{k\log(k)}+2}}$$

$$\leq \frac{\sqrt{k\log(k)}}{e}e^{\frac{2\sqrt{k}}{\sqrt{k\log(k)}+2}}$$

$$\leq e\sqrt{k\log(k)}$$

$$\leq 3\sqrt{k\log(k)} ,$$

where the first step is a re-writing, the second step uses that $\sup_{y \in \mathbb{R}} ye^{-\frac{y}{\sqrt{k\log(k)}}} \leq \sqrt{k\log(k)}/e$, and the last step uses that $\frac{2\sqrt{k}}{2+\sqrt{k\log(k)}} \leq 2$ for all $k > 10$.

The result follows by a union bound over over all three conditions. $\qquad\square$

**Lemma D.5.** *Let $T$ be a set of $n$ $\alpha$-corrupted set of points from $\mathcal{N}(\mu, I)$ according to Definition 1.1 for some $\mu$ with $\|\mu\| = O(1)$. Assume $n = \frac{k^5}{\eta^2\delta}2^{C/\epsilon^2}$ for a sufficiently large absolute constant $C$, where $k \geq 1/\epsilon^2$ and $\eta \leq \epsilon$. Denote by $z_1, \ldots, z_{\alpha n}$ the adversarial points used in Definition 1.1. Then, with probability at least $1 - \delta$, $T$ is $(\eta, \epsilon)$-good with respect to $\mu, z_1, \ldots, z_{\alpha n}$.*

*Proof.* Let $S$ be the subset of the samples that correspond to inliers and $T \setminus S$ the subset corresponding to outliers. We want to establish the two conditions of Definition 2.6. We prove each one separately:

**Proof of Item 2**   Let $\widehat{A} = \frac{1}{\alpha n}\sum_{i:x_i \in T \setminus S}\widehat{A}_i$, where $\widehat{A}_i$ as defined in Equation (1) for $\beta = \epsilon$, also let $A = \mathbf{E}[\widehat{A}]$. Now define the matrix $\Delta = \widehat{A} - A$. We will show that for each element $\Delta_{ij}$, with high probability $1 - \delta$ it holds $|\Delta_{ij}| \leq \eta/k$, which will allow us to conclude that $\|\Delta\|_{\mathrm{F}} = \sqrt{\sum_{ij}|\Delta_{ij}|^2} \leq \sqrt{k^2\eta^2/k^2} = \eta$, which in its turn will imply the desired $\|\Delta\|_{\mathrm{op}} \leq \|\Delta\|_{\mathrm{F}} \leq \eta$.

Fix $s, t \in [k]$ and denote by $e_i$ the $i$-th element of the standard basis of $\mathbb{R}^k$. By Lemma 2.3 we have that $\mathbf{E}[\Delta_{k\ell}] = 0$ and by Claim D.6 (applied with $\beta = \epsilon$), we have that

$$\mathbf{Var}[\Delta_{st}] \leq \frac{1}{(\alpha n)^2}\sum_{i:x_i \in T \setminus S}\mathbf{Var}[e_s^\top(\widehat{A}_i - A)e_t] \lesssim \frac{e^{4/\epsilon^2}d}{\alpha n} .$$

Thus, by Chebyshev's inequality, with probability at least $1 - \tau$, we have that

$$|\Delta_{st}| \leq \sqrt{\frac{\mathbf{Var}[\Delta_{st}]}{\tau}} \lesssim \sqrt{\frac{e^{4/\epsilon^2}k}{\alpha\tau n}}$$

The right hand side becomes less than $\eta/k$ when $n = O\left(\frac{k^3}{\eta^2\tau}e^{4/\epsilon^2}\right)$. Finally, we can assume $\alpha = \Omega(1)$ without loss of generality, by simply treating some of inlier points as outliers in Definition 2.6.

Now, in order to do a union bound over all pairs $(s, t)$, for $s \in [k]$, $t \in [k]$, we will use $\tau = \frac{\delta}{k^2}$ for the failure probability. This brings the final sample complexity to $n = O\left(\frac{k^5}{\eta^2\delta}e^{4/\epsilon^2}\right)$.

**Proof of Item 1**   The proof of this is a special case of the proof of Item 2 where all the $z_i$'s are the same and equal to $\mu$.

**Proof of Item 3** Let $\widehat{A}$ the sample average matrix from (1) over the whole sample set $T$ with $\beta = \epsilon$, we will show that $\operatorname{tr}(\widehat{A}) \leq 2\epsilon\sqrt{k}$ with probability $1 - \delta$. From Item 1 and Item 2 we have proved that $\left\|\widehat{A} - A\right\|_{\mathrm{F}} \leq \eta$, where $A := \frac{1}{n}\sum_{i:x_i \in T_t} z_i z_i^\top e^{-\frac{\|z_i\|^2}{\epsilon\sqrt{k}+2}}$. Consequently,

$$\operatorname{tr}(\widehat{A}) = \operatorname{tr}(A) + \operatorname{tr}(\widehat{A} - A) \leq \epsilon\sqrt{k} + \sqrt{k}\|\widehat{A} - A\|_{\mathrm{F}} \leq \epsilon\sqrt{k} + \sqrt{k}\eta \leq 2\epsilon\sqrt{k},$$

where the last inequality uses that $\eta \leq \epsilon$. $\qquad\square$

### D.1. Proof of Supporting Claims

We restate and prove the following claims that were used in the previous section.

**Claim D.6.** *Assume $k > 1/\beta^2$. Consider the random $k \times k$ matrix $\widehat{A}_x := Z_{\beta,k}F_{\beta,k}(x)$, with $x \sim \mathcal{N}(\mu, I_k)$ is $k$-dimensional normal, and $Z_{\beta,k}, F_{\beta,k}$ as defined in Equation (1). For any two unit vectors $v, u$, it holds $\mathbf{Var}[v^\top \widehat{A}_x u] \lesssim e^{4/\beta^2}\beta^2 k$.*

*Proof.* For the random variable $Y = v^\top \widehat{A}_x u$, we have that

$$\mathbf{E}[|Y|^p] = \frac{Z^p}{Z'}\int_{\mathbb{R}^k}\left|(v^\top x)(u^\top x) - \frac{\beta\sqrt{k}}{2+\beta\sqrt{k}}v^\top u\right|^p e^{-p\frac{\|x\|^2}{\beta\sqrt{k}}}e^{-\frac{\|x-\mu\|^2}{2}}dx \qquad\text{(for } Z' = (2\pi)^{k/2})$$

$$= \frac{Z^p}{Z'}\int_{\mathbb{R}^k}\left|(v^\top x)(u^\top x) - \frac{\beta\sqrt{k}}{2+\beta\sqrt{k}}v^\top u\right|^p e^{-\frac{1}{2}(\|x\|^2(\frac{2p}{\beta\sqrt{k}}+1)-2x^\top\mu+\|\mu\|^2)}dx$$

$$= \frac{Z^p}{Z'}\int_{\mathbb{R}^k}\left|(v^\top x)(u^\top x) - \frac{\beta\sqrt{k}}{2+\beta\sqrt{k}}v^\top u\right|^p e^{-\frac{c}{2}(\|x\|^2-\frac{2}{c}x^\top\mu+\frac{1}{c}\|\mu\|^2)}dx \qquad\text{(for } c = \frac{2p}{\beta\sqrt{k}}+1)$$

$$= \frac{Z^p}{Z'}\int_{\mathbb{R}^k}\left|(v^\top x)(u^\top x) - \frac{\beta\sqrt{k}}{2+\beta\sqrt{k}}v^\top u\right|^p e^{-\frac{c}{2}(\|x-\frac{1}{c}\mu\|^2+(\frac{1}{c}-\frac{1}{c^2})\|\mu\|^2)}dx$$

$$= \frac{Z^p e^{-\frac{1}{2}(1-\frac{1}{c})\|\mu\|^2}}{Z'}\int_{\mathbb{R}^k}\left|(v^\top x)(u^\top x) - \frac{\beta\sqrt{k}}{2+\beta\sqrt{k}}v^\top u\right|^p e^{-\frac{c}{2}\|x-\frac{1}{c}\mu\|^2}dx$$

$$= \frac{Z^p e^{-\frac{p}{2p+\beta\sqrt{k}}\|\mu\|^2}Z''}{Z'}\underset{x\sim\mathcal{N}(\frac{1}{c}\mu,\frac{1}{c}I)}{\mathbf{E}}\left[\left|(v^\top x)(u^\top x) - \frac{\beta\sqrt{k}}{2+\beta\sqrt{k}}v^\top u\right|^p\right] \qquad\text{(for } Z'' = (2\pi)^{k/2}\sqrt{\frac{1}{c^k}})$$

$$= \frac{Z^p e^{-\frac{p}{2p+\beta\sqrt{k}}\|\mu\|^2}}{(1+\frac{2p}{\beta\sqrt{k}})^{k/2}}\underset{x\sim\mathcal{N}(\frac{1}{c}\mu,\frac{1}{c}I)}{\mathbf{E}}\left[\left|(v^\top x)(u^\top x) - \frac{\beta\sqrt{k}}{2+\beta\sqrt{k}}v^\top u\right|^p\right].$$

For $p = 2$, this becomes

$$\mathbf{E}[|Y|^2] \lesssim \frac{(1+\frac{2}{\beta\sqrt{k}})^{k+4}e^{-\frac{2\|\mu\|^2}{4+\beta\sqrt{k}}}}{(1+4/(\beta\sqrt{k}))^{k/2}}\underset{x\sim\mathcal{N}(\frac{1}{c}\mu,\frac{1}{c}I)}{\mathbf{E}}\left[\left|(v^\top x)(u^\top x) - \frac{\beta\sqrt{k}}{2+\beta\sqrt{k}}v^\top u\right|^2\right]$$

$$\lesssim e^{4/\beta^2}e^{-\frac{2\|\mu\|^2}{4+\beta\sqrt{k}}}\underset{x\sim\mathcal{N}(\frac{1}{c}\mu,\frac{1}{c}I)}{\mathbf{E}}\left[\left|(v^\top x)(u^\top x) - \frac{\beta\sqrt{k}}{2+\beta\sqrt{k}}v^\top u\right|^2\right] \qquad\text{(explained below)}$$

$$\lesssim e^{4/\beta^2}e^{-\frac{2\|\mu\|^2}{4+\beta\sqrt{k}}}\underset{x\sim\mathcal{N}(\frac{1}{c}\mu,\frac{1}{c}I)}{\mathbf{E}}\left[\left|(v^\top x)(u^\top x)\right|^2 + \frac{\beta\sqrt{k}}{2+\beta\sqrt{k}}|v^\top u|^2\right] \qquad\text{(using } (a+b)^2 \leq 2a^2 + 2b^2)$$

$$\lesssim e^{4/\beta^2}e^{-\frac{2\|\mu\|^2}{4+\beta\sqrt{k}}}\left(\underset{x\sim\mathcal{N}(\frac{1}{c}\mu,\frac{1}{c}I)}{\mathbf{E}}\left[\left|(v^\top x)(u^\top x)\right|^2\right] + \frac{\beta\sqrt{k}}{2+\beta\sqrt{k}}\right)$$

$$= e^{4/\beta^2}e^{-\frac{2\|\mu\|^2}{4+\beta\sqrt{k}}}\left(\underset{x\sim\mathcal{N}(\frac{1}{c}\mu,\frac{1}{c}I)}{\mathbf{E}}\left[\left|(v^\top x)(av^\top x + bz^\top x)\right|^2\right] + 1\right) \qquad (u=av+bz \text{ for } z\perp v, a:=u^\top v, b:=\sqrt{1-a^2})$$

$$= e^{4/\beta^2} e^{-\frac{2\|\mu\|^2}{4+\beta\sqrt{k}}} \left( \mathop{\mathbf{E}}_{x\sim\mathcal{N}(\frac{1}{c}\mu, \frac{1}{c}I)} [a^2(v^\top x)^4 + 2ab(v^\top x)^3(z^\top x) + b^2(v^\top x)^2(z^\top x)^2] + 1 \right)$$

$$= e^{4/\beta^2} e^{-\frac{2\|\mu\|^2}{4+\beta\sqrt{k}}} \left( a^2 \mathbf{E}[x^4] + 2ab\, \mathbf{E}[x^3]\, \mathbf{E}[y] + b^2 \mathbf{E}[x^2]\, \mathbf{E}[y^2] + 1 \right) \quad (x\sim\mathcal{N}(\tfrac{1}{c}v^\top\mu, \tfrac{1}{c}) \text{ and } y\sim\mathcal{N}(\tfrac{1}{c}z^\top\mu, \tfrac{1}{c}))$$

$$\lesssim e^{4/\beta^2} e^{-\frac{2\|\mu\|^2}{4+\beta\sqrt{k}}} \left( \frac{\|\mu\|^4}{c^4} + \frac{\|\mu\|^3}{c^3} + \frac{\|\mu\|^2}{c^2} + \frac{1}{c^4} + 1 \right)$$
$$\text{(using } |a|\le 1, |b|\le 1 \text{ and } \mathbf{E}_{y\sim\mathcal{N}(z,\sigma^2)}[x^4] = z^4 + 6z^2\sigma^2 + 3\sigma^4)$$

$$\lesssim e^{4/\beta^2} e^{-\frac{2\|\mu\|^2}{4+\beta\sqrt{k}}} \left( \|\mu\|^4 + 1 \right) \qquad\qquad\qquad\qquad (\text{using } c = \tfrac{2p}{\beta\sqrt{k}} + 1 \ge 1)$$

$$\lesssim e^{4/\beta^2} \beta^2 k . \qquad\qquad\qquad\qquad\qquad (\text{using the fact } \sup_{x\in\mathbb{R}} x^4 e^{-x^2/\gamma} \le \gamma^2)$$

We now explain the second step above, which claims that $(1 + \frac{2}{\beta\sqrt{k}})^{k+4}/(1 + \frac{4}{\beta\sqrt{k}})^{k/2} \lesssim e^{4/\beta^2}$: First note that since $k \ge 1/\beta^2$ we have that $(1 + \frac{2}{\beta\sqrt{k}})^4 \le 3^4 = O(1)$, thus it suffices to prove that $(1 + \frac{2}{\beta\sqrt{k}})^k/(1 + \frac{4}{\beta\sqrt{k}})^{k/2} \lesssim e^{4/\beta^2}$. Towards this end, we will use the fact that $e^x \le (1 + x/n)^{n+x/2}$ for all $x, n > 0$. Applying this with $n = k/2$ and $x = 2\sqrt{k}/\beta$, we have that

$$e^{\frac{2\sqrt{k}}{\beta}} \le \left( 1 + \frac{4}{\beta\sqrt{k}} \right)^{\frac{k}{2} + \frac{\sqrt{k}}{\beta}} .$$

Rearranging, this gives that

$$\left( 1 + \frac{4}{\beta\sqrt{k}} \right)^{k/2} \ge e^{\frac{2\sqrt{k}}{\beta}} / \left( 1 + \frac{4}{\beta\sqrt{k}} \right)^{\sqrt{k}/\beta} . \tag{10}$$

Finally, using that, we obtain:

$$\frac{\left( 1 + \frac{2}{\beta\sqrt{k}} \right)^k}{\left( 1 + \frac{4}{\beta\sqrt{k}} \right)^{k/2}} \le \frac{\left( 1 + \frac{4}{\beta\sqrt{k}} \right)^{\sqrt{k}/\beta} \left( 1 + \frac{2}{\beta\sqrt{k}} \right)^k}{e^{\frac{2\sqrt{k}}{\beta}}} \le \frac{e^{4/\beta^2} \left( 1 + \frac{2}{\beta\sqrt{k}} \right)^k}{e^{\frac{2\sqrt{k}}{\beta}}} \le e^{\frac{4}{\beta^2} + \frac{2\sqrt{k}}{\beta} - \frac{2\sqrt{k}}{\beta}} \le e^{4/\beta^2} .$$

$$\square$$

**Claim D.7.** *Let $\mathcal{E}_i$ denote the event from* (5). *Then,*

$$\exp\left( -\frac{\|x_i\|^2 - k}{\sqrt{k}\log(k)} \right) \mathbb{1}(\mathcal{E}_i) \le \exp\left( -\frac{\|z_i\|^2}{\sqrt{k}\log(k)} + O\left( \frac{\|z_i\|\sqrt{\log(k/\eta)}}{\sqrt{k}\log(k)} \right) \right) (1/\eta)^{o(1)} .$$

*Proof.* Let $\tau := 1/n^4$, then, by the definition of the event $\mathcal{E}_i$ in (5):

$$\exp\left( -\frac{\|x_i\|^2 - k}{\sqrt{k}\log(k)} \right) \mathbb{1}(\mathcal{E}_i) \le \exp\left( \frac{-\|z_i\|^2 + O\left( \log\frac{1}{\tau} + \left( \sqrt{k} + \|z_i\| \right)\sqrt{\log(1/\tau)} \right)}{\sqrt{k}\log(k)} \right)$$

$$\le \exp\left[ -\frac{\|z_i\|^2}{\sqrt{k}\log(k)} + O\left( \frac{\log(1/\tau)}{\sqrt{k}\log(k)} + \frac{\sqrt{\log(1/\tau)}}{\sqrt{\log(k)}} + \frac{\|z_i\|\sqrt{\log(1/\tau)}}{\sqrt{k}\log(k)} \right) \right]$$

$$\le \exp\left[ -\frac{\|z_i\|^2}{\sqrt{k}\log(k)} + O\left( \frac{\log(k/\eta)}{\sqrt{k}\log(k)} + \frac{\sqrt{\log(k/\eta)}}{\sqrt{\log(k)}} + \frac{\|z_i\|\sqrt{\log(k/\eta)}}{\sqrt{k}\log(k)} \right) \right]$$

$$\le \exp\left[ -\frac{\|z_i\|^2}{\sqrt{k}\log(k)} + O\left( \frac{\log(k) + \log(1/\eta)}{\sqrt{k}\log(k)} + \frac{\sqrt{\log(k/\eta)}}{\sqrt{\log(k)}} + \frac{\|z_i\|\sqrt{\log(k/\eta)}}{\sqrt{k}\log(k)} \right) \right]$$

$$\leq \exp\left[-\frac{\|z_i\|^2}{\sqrt{k\log(k)}} + O\left(\frac{\log(1/\eta)}{\sqrt{k\log(k)}} + \frac{\sqrt{\log(k/\eta)}}{\sqrt{\log(k)}} + \frac{\|z_i\|\sqrt{\log(k/\eta)}}{\sqrt{k\log(k)}}\right)\right]$$

$$\leq \exp\left[-\frac{\|z_i\|^2}{\sqrt{k\log(k)}} + O\left(\frac{\log(1/\eta)}{\sqrt{k\log(k)}} + \sqrt{\log(1/\eta)} + \frac{\|z_i\|\sqrt{\log(k/\eta)}}{\sqrt{k\log(k)}}\right)\right]$$

$$\leq \exp\left[-\frac{\|z_i\|^2}{\sqrt{k\log(k)}} + O\left(\frac{\|z_i\|\sqrt{\log(k/\eta)}}{\sqrt{k\log(k)}}\right)\right](1/\eta)^{o(1)} .$$

In the above, we first used that $\tau = (1/n)^{O(1)}$ and $n = (k/\eta)^{O(1)}$, then we used that $\log(k)/\sqrt{k} = o(1)$. We have also used that $e^{\sqrt{\log(1/\eta)}} = (1/\eta)^{o(1)}$ and $e^{\frac{\log(1/\eta)}{\sqrt{k\log k}}} = (1/\eta)^{o(1)}$.

$\square$

**Claim D.8.** *Fix a $k > 10, 0 < \eta < 1$ and $C > 8$. The following inequalities hold:*

i. $-\frac{x^2}{\sqrt{k\log(k)}} + C\frac{x\sqrt{\log\left(\frac{k}{\eta}\right)}}{\sqrt{k\log(k)}} \leq C^2\frac{\log\left(\frac{k}{\eta}\right)}{\sqrt{k\log(k)}}$ *for all $x > 0$.*

ii. $x^2 e^{-\frac{x^2}{\sqrt{k\log(k)}} + C\frac{x\sqrt{\log\frac{k}{\eta}}}{\sqrt{k\log(k)}}} \lesssim C^2\sqrt{k\log k}\log\left(\frac{k}{\eta}\right)e^{C^2\frac{\log\left(\frac{k}{\eta}\right)}{\sqrt{k\log(k)}}}$ *for all $x > 0$.*

iii. $x^4 e^{-\frac{2x^2}{\sqrt{k\log(k)}} + C\frac{x\sqrt{\log(k/\eta)}}{\sqrt{k\log(k)}}} \lesssim C^4 k\log(k)\log^2\left(\frac{k}{\eta}\right)e^{C^2\frac{\log\left(\frac{k}{\eta}\right)}{\sqrt{k\log(k)}}}$ *for all $x > 0$.*

*Proof.* We prove each item in turn.

**Proof of i** Let $f(x) = -\frac{x^2}{\sqrt{k\log(k)}} + C\frac{x\sqrt{\log\left(\frac{k}{\eta}\right)}}{\sqrt{k\log(k)}}$ as $\lim_{x\to+\infty} f(x) = -\infty$ and $f(0) = 0$ and $f$ continuous on $(0, +\infty)$ we have that the maximum must occur on a point in $(0, +\infty)$ where the derivative is 0 either it is upper bounded than 0. Also

$$f'(x) = 0$$
$$\Rightarrow x = \frac{C}{2}\sqrt{\log\left(\frac{k}{\eta}\right)}.$$

Thus, as $f'(x) \leq \frac{C^2}{2}\frac{\log\left(\frac{k}{\eta}\right)}{\sqrt{k\log(k)}}$, we get the result.

**Proof of ii** Let $f(x) = x^2 e^{-\frac{x^2}{\sqrt{k\log(k)}} + C\frac{x\sqrt{\log\left(\frac{k}{\eta}\right)}}{\sqrt{k\log(k)}}}$. As $\lim_{x\to+\infty} f(x) = 0$ and $f(0) = 0$ and $f$ continuous on $(0, +\infty)$, we have that the maximum must occur on a point in $(0, +\infty)$ where the derivative is 0. We have

$$f'(x) = 2xe^{-\frac{x^2}{\sqrt{k\log(k)}} + C\frac{x\sqrt{\log\left(\frac{k}{\eta}\right)}}{\sqrt{k\log(k)}}} + \left(C\frac{x^2\sqrt{\log\left(\frac{k}{\eta}\right)}}{\sqrt{k\log(k)}} - \frac{2x^3}{\sqrt{k\log(k)}}\right)e^{-\frac{x^2}{\sqrt{k\log(k)}} + C\frac{x\sqrt{\log\left(\frac{k}{\eta}\right)}}{\sqrt{k\log(k)}}},$$

so solving for $x > 0$ we get $x = \frac{1}{4}\left(C\sqrt{\log\left(\frac{k}{\eta}\right)} + \sqrt{16\sqrt{k\log(k)} + C^2\log\left(\frac{k}{\eta}\right)}\right)$. Hence, it suffices to compute

$$f\left(\frac{C\sqrt{\log\left(\frac{k}{\eta}\right)} + \sqrt{C^2\log\left(\frac{k}{\eta}\right) + 16\sqrt{k\log(k)}}}{4}\right)$$

$$\leq \frac{1}{16} e^{\frac{C\sqrt{\log\left(\frac{k}{\eta}\right)}\left(C\sqrt{\log\left(\frac{k}{\eta}\right)}+\sqrt{16\sqrt{k\log(k)}+C^2\log\left(\frac{k}{\eta}\right)}\right)}{4\sqrt{k\log(k)}}} \left(C\sqrt{\log\left(\frac{k}{\eta}\right)}+\sqrt{16\sqrt{k\log(k)}+C^2\log\left(\frac{k}{\eta}\right)}\right)^2$$

$$\lesssim C^2\sqrt{k\log(k)}\log\left(\frac{k}{\eta}\right)e^{C^2\frac{\log\left(\frac{k}{\eta}\right)}{\sqrt{k\log(k)}}}.$$

Hence, $x^2 e^{-\frac{x^2}{\sqrt{k\log(k)}}+C\frac{x\sqrt{\log(k/\eta)}}{\sqrt{k\log(k)}}} \lesssim C^2\sqrt{k\log(k)}\log\left(\frac{k}{\eta}\right)e^{3C^2\frac{\log\left(\frac{k}{\eta}\right)}{\sqrt{k\log(k)}}}.$

**Proof of iii** Similarly to the previous inequality, let

$$f(x) = x^4 e^{-\frac{x^2}{\sqrt{k\log(k)}}+C\frac{x\sqrt{\log\left(\frac{k}{\eta}\right)}}{\sqrt{k\log(k)}}}.$$

As $\lim_{x\to+\infty} f(x) = -\infty$ and $f(0) = 0$ and $f$ continuous on $(0, +\infty)$, we have that the maximum must occur on a point in $(0, +\infty)$ where the derivative is $0$ either it is upper bounded than $0$. Also, we have that

$$f'(x) = 0$$

$$\Rightarrow \left(4x^3 - \frac{2x^5}{\sqrt{k\log(k)}} + C\frac{x^4\sqrt{\log\left(\frac{k}{\eta}\right)}}{\sqrt{k\log(k)}}\right)e^{-\frac{x^2}{\sqrt{k\log(k)}}+C\frac{x\sqrt{\log\left(\frac{k}{\eta}\right)}}{\sqrt{k\log(k)}}} = 0$$

$$\Rightarrow x = \frac{1}{8}\left(C\sqrt{\log\left(\frac{k}{\eta}\right)} + \sqrt{64\sqrt{k\log(k)}+C^2\log\left(\frac{k}{\eta}\right)}\right). \qquad\text{(for } x > 0)$$

Moreover,

$$f\left(\frac{1}{8}\left(C\sqrt{\log\left(\frac{k}{\eta}\right)} + \sqrt{64\sqrt{k\log(k)}+C^2\log\left(\frac{k}{\eta}\right)}\right)\right)$$

$$\lesssim C^4 k\log(k)\log^2\left(\frac{k}{\eta}\right)e^{C^2\frac{\log\left(\frac{k}{\eta}\right)}{\sqrt{k\log(k)}}}.$$

$\square$

# E. Full Proof of Theorem 1.2

We now combine everything to prove the main theorem, which we restate below:

**Theorem 1.2.** (Main Algorithmic Result) *Let $d \in \mathbb{Z}_+$ denote the dimension, $\mu \in \mathbb{R}^d$ be an unknown mean vector, $\epsilon \in (0, 1)$ be an accuracy parameter, and $\alpha \leq 0.49$ be a contamination parameter. There exists an algorithm that takes as input $\epsilon$, draws $n = \tilde{O}(d/\epsilon^{2+o(1)} + 2^{O(1/\epsilon^2)})$ $\alpha$-corrupted samples from $\mathcal{N}(\mu, I)$ under the mean-shift model (Definition 1.1), runs in $\mathrm{poly}(n, d)$ time, and outputs $\hat{\mu}$ such that with probability at least $0.99$ it holds $\|\hat{\mu} - \mu\| \leq \epsilon$.*

*Proof.* The main while loop of the algorithm maintains a subspace $\mathcal{V}_t$ whose dimension, denoted by $k$ in the pseudocode, starts from $d$ and can only decrease from a round to the next one. We will examine two distinct "phases" (or parts) of this while loop: Phase 1 will refer to all the iterations during which $C\log^4(d)/\epsilon^5 \leq k \leq d$, and phase 2 will refer to all the iterations with $1/\epsilon^2 \leq k \leq C\log^4(d)/\epsilon^5$. The algorithm and the analysis will be slightly different for each phase. Regarding notation, let us denote by $T_1$ the number of rounds of phase 1 and by $T_2$ the number of rounds of phase 2 (i.e., phase 1 consists of all the iterations for $t = 1, \ldots, T_1$ and phase 2 consists of the iterations for $t = T_1 + 1, \ldots, T_1 + T_2$). The proof of correctness relies on the following claims regarding each part of the algorithm. We first state the claims, we then show how we can prove the theorem using the claims, and we finally prove each claim individually.

1. **Warm start**: If $\widehat{\mu}_0$ denotes the estimator from line 1 of the algorithm, it holds $\|\widehat{\mu}_0 - \mu\| = O(1)$ with probability 0.999.

2. **Phase 1** $(C\log^4(d)/\epsilon^5 \leq k \leq d)$:

   (a) Let $\mathcal{E}_t$ for $t \in 1, 2, \ldots$ be the event that the set $T_t$ from line 1, after the transformation of line 1 is $(\eta_t, \sqrt{\log(k)})$-good (cf. Definition 2.6) with respect to $\tilde{\mu}_t, \tilde{z}_1^{(t)}, \ldots, \tilde{z}_\alpha^{(t)}$, where $\eta_t$ is the parameter set in line 1, $\tilde{\mu}_t = \mathrm{Proj}_{\mathcal{V}_t}(\mu - \widehat{\mu}_0)$ and $\tilde{z}_1^{(t)}, \ldots, \tilde{z}_\alpha^{(t)}$ are some vectors in $\mathcal{V}_t$. Then, with probability at least 0.999, the events $\mathcal{E}_t$ for $t = 0, 1, \ldots, \log d$ are all true.

   (b) Assuming $\mathcal{E}_t$ hold for $t = 1, \ldots, T_1$, Phase 1 terminates after at most $T_1 \leq \log d$ iterations.

   (c) Let $C$ be a sufficiently large absolute constant. Denote by $\mathcal{V}_{t+1}$ the same subspace as in line 1 of the algorithm and by $\mathcal{V}_{t+1}^\perp$ its orthogonal complement. During the $t$-th iteration of the while loop, when the execution reaches line 1, it holds $\|\mathrm{Proj}_{\mathcal{V}_{t+1}^\perp}(\widehat{\mu}_0 - \mu)\| \leq C\sum_{t'=1}^{t}\sqrt{\eta_{t'}}$, where $\eta_{t'}$ is the value set in line 1.

3. **Phase 2** $(1/\epsilon^2 \leq k \leq C\log^4(d)/\epsilon^5)$:

   (a) Let $\mathcal{E}_t'$ be the event that the set $T_t$ from line 1, after the transformation of line 1 is $(\eta_t, \epsilon)$-good (cf. Definition 2.6) with respect to $\tilde{\mu}_t, \tilde{z}_1^{(t)}, \ldots, \tilde{z}_\alpha^{(t)}$, where $\eta_t$ is the parameter set in 1, $\tilde{\mu} = \mathrm{Proj}_{\mathcal{V}_t}(\mu - \widehat{\mu}_0)$ and $\tilde{z}_1^{(t)}, \ldots, \tilde{z}_\alpha^{(t)}$ are some vectors in $\mathcal{V}_t$. Then, with probability at least 0.999, the events $\mathcal{E}_t'$ for $t = T_1 + 1, \ldots, T_1 + 100\log(\log(d)/\epsilon)$ are all true.

   (b) Assuming the events $\mathcal{E}_t'$ from above hold, in every iteration of Phase 2 the dimension $k$ gets halved. As a corollary, the number of iterations of Phase 2 (i.e., number of iterations for which $1/\epsilon^2 \leq k \leq C\log^4(d)/\epsilon^5$) is $T_2 \leq 100\log(\log(d)/\epsilon)$.

   (c) Let $C$ be a sufficiently large absolute constant. Denote by $\mathcal{V}_{t+1}$ the same subspace as in line 1 of the algorithm and by $\mathcal{V}_{t+1}^\perp$ its orthogonal complement. Assume that the events $\mathcal{E}_t$ for $t \in [t_{\max}]$ are all true. Then, the following holds for any $t \in [t_{\max}]$: During the $t$-th iteration of the while loop, when the execution reaches line 1, it holds $\|\mathrm{Proj}_{\mathcal{V}_{t+1}^\perp}(\widehat{\mu}_0 - \mu)\| \leq C\sum_{t'=1}^{t}\sqrt{\eta_{t'}}$, where $\eta_{t'}$ are the values set in lines 1 and 1.

4. **Estimator for remaining subspace**: With probability at least 0.999, the lines 1-1 of the algorithm find a vector $\widehat{\mu}_1 \in \mathcal{V}_t$ such that $\|\widehat{\mu}_1 - \mathrm{Proj}_{\mathcal{V}_t}(\mu)\| \leq \epsilon$.

We now show how given the claims above it follows that, with probability at least 0.99, the output of the algorithm $\widehat{\mu} := \widehat{\mu}_0 + \widehat{\mu}_1$ satisfies $\|\widehat{\mu} - \mu\| \leq \epsilon$. Without loss of generality, it suffices to show $\|\widehat{\mu} - \mu\| = O(\epsilon)$ as, if this is true, then one can also obtain error exactly $\epsilon$ by running the algorithm with $c\epsilon$ in place of $\epsilon$, for $c$ being a sufficiently small constant. Let $t = T_1 + T_2$, so that $\mathcal{V}_t$ denotes the subspace after exiting the while loop. By decomposing the true mean into the projections onto the two orthogonal subspaces we have that $\widehat{\mu} = \mathrm{Proj}_{\mathcal{V}_t^\perp}(\mu) + \mathrm{Proj}_{\mathcal{V}_t}(\mu)$: By the Pythagorean theorem:

$$\|\widehat{\mu} - \mu\|^2 = \left\|\mathrm{Proj}_{\mathcal{V}_t^\perp}(\widehat{\mu}_0 - \mu) + \mathrm{Proj}_{\mathcal{V}_t}(\widehat{\mu}_1 - \mu)\right\|^2 = \left\|\mathrm{Proj}_{\mathcal{V}_t^\perp}(\widehat{\mu}_0 - \mu)\right\|^2 + \left\|\mathrm{Proj}_{\mathcal{V}_t}(\widehat{\mu}_1 - \mu)\right\|^2 .$$

The last term is $\left\|\mathrm{Proj}_{\mathcal{V}_t}(\widehat{\mu}_1 - \mu)\right\| \leq \epsilon$ by Item 4. It suffices to bound the first term. Towards this end, denote $D := C\log^4(d)/\epsilon^5$, which is the dimension during the first iteration of phase 2. Item 2c for $t = T_1 + T_2$ (which according to our notation denotes the last iteration of Phase 2) yields the following (we explain the derivations below):

$$\left\|\mathrm{Proj}_{\mathcal{V}_t^\perp}(\widehat{\mu}_0 - \mu)\right\| \lesssim \sum_{t'=1}^{T_1+T_2}\sqrt{\eta_{t'}} \tag{11}$$

$$\leq \sum_{t'=1}^{T_1}\sqrt{\eta_{t'}} + \sum_{t'=T_1+1}^{T_1+T_2}\sqrt{\eta_{t'}} \tag{12}$$

$$\leq T_1\frac{\epsilon}{\log d} + \sum_{t'=T_1+1}^{T_1+T_2}\sqrt{\eta_{t'}} \tag{13}$$

$$\leq \epsilon + \sum_{t'=T_1+1}^{T_1+T_2} \sqrt{\eta_{t'}} \qquad\qquad (T_1 \leq \log d \text{ by Item 2b})$$

$$\leq \epsilon + \left( \sqrt{\frac{36\epsilon}{\sqrt{D}}} + \sqrt{\frac{36\epsilon}{\sqrt{D/2}}} + \cdots + \sqrt{\frac{36\epsilon}{\sqrt{1/\epsilon^2}}} \right) \tag{14}$$

$$\leq \epsilon + \frac{6\sqrt{\epsilon}}{D^{1/4}} \sum_{i=0}^{\lg(D\epsilon^2)} 2^{i/4} \tag{15}$$

$$\leq \epsilon + \frac{6\sqrt{\epsilon}}{(2^{1/4}-1)D^{1/4}} 2^{\frac{\lg(D\epsilon^2)}{4}} \tag{16}$$

$$= \epsilon + \frac{6\sqrt{\epsilon}}{(2^{1/4}-1)D^{1/4}} (D\epsilon^2)^{1/4} \lesssim \epsilon . \tag{17}$$

We proceed to explain the steps above: (14) uses the definition of $\eta_t := 36\epsilon/\sqrt{k}$. The dimension $k$ for the first round of Phase 2 is $D$ and in every subsequent round it gets divided by 2 (by the claim of Item 3b). Phase 2 ends when the dimension becomes $1/\epsilon^2$, which corresponds to the last term in the series. (15) is a rewriting. The summation goes from $i = 0$ to $\lg(D\epsilon^2)$ because $\lg(D\epsilon^2)$ is the solution to the equation $D/2^x = 1/\epsilon^2$ which seeks to determine after how many rounds of halving the dimension becomes $1/\epsilon^2$. The next line, (16) uses the closed form formula for that series.

We now prove all the individual claims.

**Proof of Item 1**   This holds with probability at least 0.999 by an application of Corollary 2.12 in Diakonikolas & Kane (2023). Without loss of generality, we apply this corollary with the fraction of outliers being $\alpha = \Omega(1)$ since we can always treat some of the inliers as outliers in the model of Definition 1.1. That corollary yields that $n_0 = O(d)$ samples suffice.

**Proof of Item 2b**   We claim that each iteration of Phase 1 of the while loop decreases the dimension from $k$ to $k' := 18\sqrt{k\log(k)}/\eta_t$. This is because $\mathcal{E}_i$ hold for $i = 1, \ldots, t$, and Item 3 of Definition 1.1 states that $\operatorname{tr}(\widehat{A}) \leq 18\sqrt{k\log(k)}$. This means that in line 1 of the algorithm, the number of eigenvectors with eigenvalue greater than $\eta_t$ can be at most $\operatorname{tr}(\widehat{A}_t)/\eta_t \leq 18\sqrt{k\log(k)}/\eta_t$.

The dimension thus gets divided by 2 whenever $18\sqrt{k\log(k)}/\eta_t < k/2$. By plugging in the value $\eta_t := (\epsilon/\log d)^2$ we obtain that the dimension gets halved whenever $k > 36^2 \cdot \log^5(d)/\epsilon^4$. This indeed holds because of line 1. Hence the number of iterations is at most $\log(d)$.

**Proof of Item 2a**   During the $t$-th round of our algorithm let $z_i^{(t)}$ denote the outlier centers that the adversary chooses for the samples, in the model of Definition 1.1. Note that performing the mean shift $\tilde{x}_i = x_i - \widehat{\mu}_0$, means that the shifted points $\tilde{x}_i$ effectively come from the model described in Definition 1.1 with shifted mean $\tilde{\mu} = \mu - \widehat{\mu}_0$ and shifted outlier centers $\tilde{z}_i^{(t)} = z_i^{(t)} - \widehat{\mu}_0$. Similarly, the projection operation of line 1 makes the points essentially come from a model that uses mean $\tilde{\mu}_t = \operatorname{Proj}_{\mathcal{V}_t}(\mu - \widehat{\mu}_0)$.

Now by a union bound over all of the $\log d$ iterations of Phase 1 and Lemma 2.7 with $\eta = \eta_t = (\epsilon/\log d)^2$ and $\delta = 10^{-3}/\log d$, we have that all of the sets $T_t$ of points drawn in line 1 will be $(\eta_t, \sqrt{\log(k)})$-good with probability at least 0.999. Note that the condition $\|\tilde{\mu}_t\| \leq \|\mu - \widehat{\mu}_0\| = O(1)$ that this lemma requires has already been established in Item 1, thus the lemma is indeed applicable. The sample complexity that this lemma yields is $k\log^3 k (1/\eta)^{2+o(1)}/\delta$. By using $\eta = \eta_t = (\epsilon/\log d)^2$ and $\delta = 10^{-3}/\log d$ this becomes $k(\log(d)/\epsilon)^{2+o(1)}$ which is smaller than the number of samples $n_1$ that we use in our algorithm (cf. line 1).

**Proof of Item 2c**   Proof by induction. Denote by $C$ the constant in the statement of Lemma 2.9. The base case $t = 1$ follows immediately by Lemma 2.9 applied with $\mathcal{U} = \mathbb{R}^d$ and $\mathcal{V} = \mathcal{V}_2^{\perp}$. For the induction step, assume that the claim

holds for $t$ and we will show it for $t+1$. Let $\mathcal{V}_{t+1}$ denote the subspace maintained by the algorithm at the end of (line 1) the $t$-th iteration. By inductive hypothesis, we know that $\|\text{Proj}_{\mathcal{V}_{t+1}^\perp}(\widehat{\mu}_0 - \mu)\| \le C \sum_{\tau=1}^{t} \sqrt{\eta_\tau}$. The goal is to show the bound for the end of the next iteration, i.e., show that $\|\text{Proj}_{\mathcal{V}_{t+2}^\perp}(\widehat{\mu}_0 - \mu)\| \le C \sum_{\tau=1}^{t+1} \sqrt{\eta_\tau}$. We split $\mathcal{V}_{t+2}^\perp$ into two parts: $\mathcal{V}_{t+2}^\perp \cap \mathcal{V}_{t+1}$ and $\mathcal{V}_{t+2}^\perp \cap \mathcal{V}_{t+1}^\perp$. We now apply Lemma 2.9 with $\mathcal{U} = \mathcal{V}_{t+1}$, $\beta = \sqrt{\log(d)}$ and $\mathcal{V} = \mathcal{V}_{t+2}^\perp \cap \mathcal{V}_{t+1}$. The lemma is indeed applicable because the set $T_t$ of points are $(\eta_t, \sqrt{\log(k)})$-good. The application of the lemma yields that $\|\text{Proj}_{\mathcal{V}_{t+2}^\perp \cap \mathcal{V}_{t+1}}(\widehat{\mu}_0 - \mu)\| \le C\sqrt{\eta_{t+1}}$. Also, the inductive hypothesis that we mentioned earlier implies that $\|\text{Proj}_{\mathcal{V}_{t+2}^\perp \cap \mathcal{V}_{t+1}}(\widehat{\mu}_0 - \mu)\| \le C \sum_{\tau=1}^{t} \sqrt{\eta_\tau}$. Combining the previous two we have that $\|\text{Proj}_{\mathcal{V}_{t+2}^\perp}(\widehat{\mu}_0 - \mu)\| \le C \sum_{\tau=1}^{t+1} \sqrt{\eta_\tau}$.

**Proof of Item 3a** This follows by an application of Lemma 2.8 with probability of failure $\delta = 10^{-5}/\log(\log(d)/\epsilon)$. Note that the requirement of that lemma that $k > 1/\epsilon^2$ holds since the algorithm would exit entirely the while loop of line 1 otherwise. Since the dimension is $k \le C \log^4(d)/\epsilon^5$ during this phase of the algorithm, and $\eta_t = O(\epsilon/\sqrt{k})$, the sample complexity $n = \frac{k^5}{\eta_t^2 \delta} 2^{O(1/\epsilon^2)}$ mentioned in that lemma is overall $2^{O(1/\epsilon^2)} \text{polylog}(d)$. By a union bound over the iterations of the algorithm, we have that all the sets $T_t$ during the first $100 \log(\log(d)/\epsilon)$ rounds of Phase 2 will be $(\eta_t, \epsilon)$-good (and as we will show below, Phase 2 will not have more than $100 \log(\log(d)/\epsilon)$-many rounds).

**Proof of Item 3b** Consider a single iteration of Phase 2 of the algorithm, and let $\widehat{A}_t$ be the matrix from line 1. Since the set of points on which $\widehat{A}_t$ is computed is $(\eta_t, \epsilon)$-good, we have that $\text{tr}(\widehat{A})$ is at most $18\epsilon\sqrt{k}$ (cf. Item 3 of Definition 2.6). If $k'$ denotes the number of eigenvalues larger than $\eta_t$ then we have $k' \le 18\epsilon\sqrt{k}/\eta_t$. Note that, by definition of $\eta_t := 36\epsilon/\sqrt{k}$,

$$\frac{18\epsilon\sqrt{k}}{\eta_t} = \frac{18\epsilon\sqrt{k}\sqrt{k}}{36\epsilon} \le k/2.$$

The dimension in each round is therefore being halved.

**Proof of Item 3c** This follows by the same argument we used for Phase 1, but applying Lemma 2.9 with $\beta = \epsilon$ instead of $\beta = \sqrt{\log k}$.

**Proof of Item 4** This follows by an application of Proposition 2.1.

**Runtime and Sample Complexity of Algorithm 1** Let $n_0, n_1, n_2$ be as defined in line Algorithm 1 of the algorithm. Recall that we denote by $T_1$ the number of iterations of the first phase of the while loop (interactions for which $C \log^4(d)/\epsilon^5 \le k \le d$) and by $T_2$ the number of the remaining iterations of the while loop. The sample complexity of the algorithm is the following:

$$n = n_0 + n_1 \cdot T_1 + n_2 \cdot (T_2 + 1)$$
$$= O(d) + \frac{d \, \text{polylog}(d)}{\epsilon^{2+o(1)}} + 2^{O(1/\epsilon^2)} \text{polylog}(d)$$
$$= \frac{d \, \text{polylog}(d)}{\epsilon^{2+o(1)}} + 2^{O(1/\epsilon^2)} \text{polylog}(d).$$

Regarding runtime: The runtime of the warm-start step is $\tau_1 = \text{poly}(n_0 d)$. The runtime of the while-loop part of the algorithm is $\tau_2 = \text{poly}(n_1 d)$ because each iteration runs in polynomial time and we have at most $T_1 + T_2 = O(\log d)$ iterations. The runtime of the last step of the algorithm (1) is $\tau_3 = 2^{O(k)} \text{poly}(n_2 d)$, where $k = 1/\epsilon^2$ here denotes the dimension of the subspace $\mathcal{V}_t$ for $t = T_1 + T_2$, i.e., the dimension that we end up after the while loop finishes. Since $n_2 = 2^{\Theta(1/\epsilon^2)}$, that runtime is overall $\tau_3 = \text{poly}(n_2 d)$. Thus, the overall runtime of the algorithm is polynomial in the size of the input.

$\square$

## F. Higher Breakdown Point and Adaptivity

In this section, we use Lepskii's method (Lepskii, 1991; Birgé, 2001) to prove Theorem F.1 (stated below), a generalization of Theorem 1.2 which provides similar guarantees as Theorem 1.2 but works for any contamination parameter $\alpha$, and does not require a priori knowledge of $\alpha$. Specifically, we can generalize the algorithm to work for $\alpha \in (0, 1/2 - c)$ for any $c < 1/2$, which is unknown to the algorithm. The price that we pay for that is a slightly larger dependence on $\epsilon$ in the sample complexity and the fact that the error becomes $O_c(\epsilon)$ instead of $\epsilon$, where the $O_c$ notation hides factors that depend on $c$ (i.e., if $c$ is a constant, the error increases only by a constant factor). Moreover, even in one dimension, Kotekal & Gao (2025) has shown that consistent estimation with arbitrarily small error $\epsilon$ using the information theoretic optimal of $2^{\Theta(1/\epsilon^2)}$ samples is only possible when $c > 2^{-\Theta(1/\epsilon^2)}$, thus we only consider that regime in this section.

**Theorem F.1.** (Higher breakdown point) *Let $d \in \mathbb{Z}_+$ denote the dimension, $\mu \in \mathbb{R}^d$ be an unknown mean vector, $\epsilon \in (0, 1)$ be an accuracy parameter. Fix $n = (d \cdot (1/\epsilon)^C + 2^{C/\epsilon^2}) \log^C(d)$, for a sufficiently large constant $C$. Suppose that we have sample access to $\alpha$-corrupted samples from $\mathcal{N}(\mu, I)$ under the mean-shift model (Definition 1.1) with contamination parameter $\alpha \in (0, 1/2 - c)$ where $2^{-1/(2\epsilon^2)} < c \leq 1/2$. There exists an algorithm that takes as input $\epsilon$, draws $n$ samples, runs in $\mathrm{poly}(n, d)$ time, and outputs $\widehat{\mu}$ such that with probability at least $0.99$ it holds $\|\widehat{\mu} - \mu\| \leq O_c(\epsilon)$.*

The first step towards proving Theorem F.1 is to show that there exists an estimator that, *having $\gamma$ as input* achieves error $O_\gamma(\epsilon)$, i.e., an estimator that requires knowledge of $\gamma$ but $\gamma$ can be arbitrary. We show this in Claim F.2. Then, we can use Lepskii's method to obtain an estimator with the same error, but without knowledge of $\gamma$.

**Claim F.2.** *Let $d \in \mathbb{Z}_+$ denote the dimension, $\mu \in \mathbb{R}^d$ be an unknown mean vector and $\epsilon \in (0, 1)$ be an accuracy parameter. Let $\alpha$ be a contamination parameter. There exists an algorithm that takes as input $\epsilon$, $\alpha$, draws $n = \tilde{O}(d/(\delta\epsilon^{2+o(1)}) + 2^{O(1/\epsilon^2)}/\delta)$ $\alpha$-corrupted samples from $\mathcal{N}(\mu, I)$ under the mean-shift model (Definition 1.1), runs in $\mathrm{poly}(n, d)$ time, and outputs $\widehat{\mu}$ such that, if $\alpha \leq 1/2 - 2^{-1/(2\epsilon^2)}$, then with probability at least $1 - \delta$ it holds $\|\widehat{\mu} - \mu\| \leq e^{O(1/(1-2\alpha)^2)}\epsilon$.*

*Proof.* For notational convenience, instead of working with the contamination parameter $\alpha$ from Definition 1.1, we will use the reparameterization $\gamma = \frac{1}{1-2\alpha}$. We now describe the modifications that we need to do to Algorithm 1 and its analysis in order to obtain Claim F.2

First, we replace the rough estimation step of line 1 in Algorithm 1 with an estimator that takes $\alpha$ in its input and achieves error $O(\gamma)$ instead of an $O(1)$ (e.g., Dalalyan & Minasyan (2022), Exercise 2.10 Diakonikolas & Kane (2023)).

This change affects the error given by the analysis of the dimensionality reduction, Lemma 2.9. It is easy to see that the conclusion of that lemma will now become $\|\mathrm{Proj}_\mathcal{U}(\mu)\| = O(e^{\gamma^2}\sqrt{\eta})$ instead of $\|\mathrm{Proj}_\mathcal{U}(\mu)\| = O(e^{\gamma^2}\sqrt{\eta})$. This is because in the last line of the proof of Lemma 2.9 we have that for any $v$ such that $v^\top \widehat{A} v \leq \eta$

$$(v^\top \mu)^2 \leq 2\eta \, e^{\frac{\|\mu\|^2}{\beta\sqrt{k}+2}} \lesssim e^{\gamma^2}\eta.$$

As a result, we have that $\|\mathrm{Proj}_\mathcal{U}(\mu)\| = e^{O(\gamma^2)}\sqrt{\eta}$ for $\mathcal{U}$ the subspace of $V_t$ such that $v^\top \widehat{A} v \leq \eta$ for all unit vectors $v \in \mathcal{U}$. So propagating this change in the analysis of Algorithm 1 in Appendix E we get the $e^{O(\gamma^2)}\epsilon$ error will appear as the contribution to the error by the subspace $\mathcal{V}_t^\perp$.

For the error on the orthogonal subspace, we need to calculate the error that our inefficient estimator in the final step of our algorithm will attain. Note that from Kotekal & Gao (2025) we have that when using $2^{O(1/\epsilon^2)}$ samples the one dimensional estimator achieves error $\frac{1}{\sqrt{\log(1+2^{-O(\epsilon^2)/\gamma^2})}} \leq e^{O(\gamma^2)}\epsilon$ when $\gamma \leq 2^{1/(2\epsilon^2)}/2$. As a result, a more refined version of Proposition 2.1 exists error $\widehat{\mu}$ is $\|\widehat{\mu} - \mu\| \leq e^{O(\gamma^2)}\epsilon$ with probability $0.99$.

Finally, although our analysis in Theorem 1.2 provides a result that holds with a constant probability, this was done only for simplicity. It is easy to verify that because the conclusion of Lemmas 2.7 and 2.8 holds with probability $1 - \delta$ with $1/\delta$ blowup in the sample complexity, we can follow the analysis done in Appendix E to obtain a high probability conclusion for the final error of the algorithm. $\square$

Now we will use Claim F.2 along with Lepskii's method to get an error guarantee that depends on the true parameter $\gamma$ without it being known to the algorithm. We state Lepskii's method guarantee below and then prove Theorem F.1.

**Fact F.3** (Lepskii's Method). *Let $\mu \in \mathbb{R}^d$, $A, B > 0$, $\gamma \in [A, B]$, and a non-decreasing function $r : \mathbb{R}^+ \to \mathbb{R}^+$. Suppose $Alg(\gamma')$ is a black-box algorithm that is guaranteed to return a vector $\widehat{\mu}$ such that $\|\widehat{\mu} - \mu\|_2 \leq r(\gamma')$, with probability at least $1 - \delta$, whenever $\gamma' \geq \gamma$. Then, there exists an algorithm that returns $\widehat{\mu}'$ such that, with probability at least $1 - O(\log(B/A))\delta$, it holds $\|\widehat{\mu}' - \mu\|_2 \leq 3r(2\gamma)$. Moreover, this algorithm calls Alg at most $O(\log(B/A))$ times.*

*Proof of Theorem F.1.* We will apply Lepskii's method (cf. Fact F.3), while using the algorithm of Claim F.2 with $\delta = 0.01\epsilon^2/\log(d)$ for Alg and $r(\gamma) := e^{O(\gamma^2)}\epsilon$. By assumption $\alpha \leq 1/2 - 1/\sqrt{n}$ (and $\alpha \geq 1/n$ otherwise we can treat one of the samples as outlier to make it at least $1/n$) thus $\gamma$ will belong in the interval $[n/(n-2), \sqrt{n}/2)$ thus Lepskii's method will search over the interval $[A, B]$ where $A = n/(n-2)$ and $B = \sqrt{n}/2$.

Now if we use the algorithm from Claim F.2 for this fixed $n$ in Lepskii's method the number of calls to our algorithm will be at most $\log(\sqrt{n}/(n/(n-2))) \leq \log(\sqrt{n}) \leq \log(d2^{O(1/\epsilon^2)}) \leq \log(d)/\epsilon^2$ (this is why we used Claim F.2 with $\delta = 0.01\epsilon^2/\log(d)$ in the beginning). Hence by the union bound, the probability of failure of any call of the algorithm in Lepskii's method is at most $\log(d)/\epsilon^2\delta \leq 0.01$. Therefore, by Fact F.3, we obtain that there exists an algorithm that returns an estimate $\widehat{\mu}'$ such that $\|\widehat{\mu}' - \mu\|_2 \leq 3r(2\gamma) \leq e^{O(\gamma^2)}\epsilon = O_\gamma(\epsilon)$ with probability at least $0.99$.

Regarding the sample complexity of our algorithm, as at each call we use the same number of samples $n$ and the number of such calls is at most $\log(d)/\epsilon^2$, we have that the total number of samples is at most $(\log(d)/\epsilon^2)n \leq \tilde{O}(d \operatorname{poly}(1/\epsilon) + 2^{O(1/\epsilon^2)})$. Furthermore, the runtime of the algorithm is $(\log(d)/\epsilon^2) \operatorname{poly}(n, d) = \operatorname{poly}(n, d)$, as Lepskii's method consists of $\log(d)/\epsilon^2$ calls to the algorithm of Claim F.2. $\qquad\square$

