# OpenReview forum: "Efficient Multivariate Robust Mean Estimation Under Mean-Shift Contamination"
_ICML.cc/2025/Conference — ICML 2025 poster_

### Official Review · Reviewer_HXFc · 2025-03-06

**Overall Recommendation:** 3

**Summary:**

In this paper the authors look at the problem of robust mean estimation under the mean shift contamination. Here one receives samples from a standard d-variate Gaussian with mean m such that m is $\mu$ with 2/3 probability and can be something else otherwise. The goal is to recover $\mu$ up to an $\epsilon$ error in L2 norm. Such a guarantee is information theoretically impossible under the Huber's contamination model. However, in this case one can use the fact that the contamination is not arbitrary but only a mean shift to solve the problem.

**Claims And Evidence:**

The authors achieve a sample complexity of $poly(d,2^{\epsilon^{-2}})$ and a time complexity of $poly(n,d)$ to solve this problem.

The claims look clear and convincing to me.

**Essential References Not Discussed:**

NA

**Experimental Designs Or Analyses:**

However, any experimental evaluation is missing. I thought experimental verification of the proposed algorithm would have added further value to the main claims of the paper.

**Methods And Evaluation Criteria:**

The authors give sound theoretical arguments to justify their paper

**Other Comments Or Suggestions:**

NA

**Other Strengths And Weaknesses:**

Strengths: as mentioned before one of the main contribution is in finding out this problem where the recently proposed spectral techniques could be applied.

Weakness: the main weakness I see that is the result looks incremental. As I said previously, this particular algorithm (spectral filtering algorithm) and its analysis have been investigated so many times in recent years. Also, lack of experimental evidence is another weakness of the paper given that it is easy to experiment with synthetic Gaussian data. This makes me less excited about the paper for a top ML venue such as ICML.

**Questions For Authors:**

Where is the dependence on $\alpha$ hiding in Theorem 1.2? I would guess the problem becomes harder as $\alpha$ gets larger.

**Relation To Broader Scientific Literature:**

I think this particular spectral filtering algorithm and its variation has been extensively looked at numerous results in the last decade. This paper builds on this technique and one of the main contribution is finding out that this problem of mean estimation under mean-shift contamination where the aforesaid technique can be applied.

**Theoretical Claims:**

I haven't gone through the theoretical proofs given in the paper. But based on the overview given in 1.2 and the prior results on this topic, the results look correct to me.

The main idea is to find a direction where the mean has a large projection and once found it suffices to run an one-dimensional estimation along that direction. Along the way, the authors manage to address several technical hurdles, in particular, while obtaining the aforesaid direction.

---

> ### Author Rebuttal · Authors · 2025-04-01
>
> We thank the reviewer for their effort and time in assessing our work. We respond to the points raised individually below:
>
> (**Experiments**) We would like to emphasize that our primary contribution is to characterize the computational-statistical landscape for this fundamental learning task—in terms of error guarantee, sample complexity, and runtime. That being said, our algorithm is already fairly simple and potentially practical, as it uses a dimension-reduction procedure with each step essentially being an SVD of a reweighted covariance matrix. Also, while it is easy to generate synthetic Gaussian data, it is not clear what distribution on errors should be considered, making the meaningfulness of experiments questionable.
>
> (**Novelty**) We would like to stress that our work develops a novel spectral dimension-reduction technique, rather than applying/adapting methods from prior work. To the best of our knowledge, there are two “spectral methods” proposed in prior work for robust mean estimation (in Huber’s model). Both of them are significantly different than our algorithm, as we discuss below:
>
>
> * (*Spectral filtering from Diakonikolas et. al (2016)*) This method consists of iteratively computing the top eigenvector of the empirical covariance and removing points whose projected deviation from the mean along that direction is higher than a carefully chosen threshold. This has the effect of removing more inliers than outliers until the dataset becomes so clean that the empirical mean is accurate. Importantly, this iterative filtering approach cannot yield consistent estimation (for the mean-shift model that we study), essentially because it does not take into account the additional structure of the outliers.
> The algorithm we propose is fundamentally different. Rather than removing points based solely on the top eigenvector of the empirical covariance, our approach performs dimension reduction. It uses multiple eigenvectors of a reweighted covariance matrix as the subspace of the next iteration. In our method, each point is assigned an exponential weight, with carefully designed reweighting factors that are updated throughout the execution of the algorithm (see lines 126-144, second column and 175-186, first column for intuition on choosing the reweighting factor). Finally, once the problem’s dimension has been reduced sufficiently, we employ an inefficient estimator on the lower-dimensional data.
>
>
>
> * (*Dimension reduction from Lai et al. (2016)*) Although the goal in that technique is also to reduce the dimension, as we explain in lines 662-671 in Appendix A of the paper and in our response to Reviewer BKkQ, the similarities are only superficial. The method from Lai et al. (2016) uses the standard covariance matrix to identify the subspace for the next iteration. This method also fails to leverage the special nature of the outliers. In contrast, our method reweights the space with an exponential function that encodes our prior knowledge on the outliers in order to truncate their effect in the second moment matrix. Although one could broadly view this reweighting as a kind of “soft filtering”, this is fundamentally different from the filtering of the previous paragraph: there, points were removed based solely on their deviation along a single direction, where here the weight assigned to each point is based on the norm of the vector. Moreover, as we comment in lines 662-671 of the paper and in our response to Reviewer BKkQ, our dimension reduction comes with a number of technical obstacles that are due to the nature of our reweighting and need to be carefully addressed.
>
> (**Dependence on $\alpha$**) Theorem 1.2 holds for any $\alpha \leq 0.49$. As we reply to Reviewer oS19, the most interesting parameter regime in our opinion  is when the fraction of outliers $\alpha$ is a positive constant. Essentially,  this is the hardest setting for efficient estimation. In that regime, no sub-exponential time algorithm was known to achieve error lower than $\Omega(1)$, while our algorithm can obtain arbitrarily small error in sample-polynomial time; and its sample complexity is information-theoretically near-optimal (as follows from the lower bound of Kotekal & Gao, 2025). As we show in Appendix F, the parameter $0.49$ in the condition of Theorem 1.2 can be replaced by any other absolute constant strictly smaller than $½$ (and the error will degrade by an absolute constant factor). Furthermore, if one prefers to use the reparameterization $\alpha = ½ - c$, where $c$ is a parameter (rather than an absolute constant), the error degrades in a way that increases as $c$ approaches zero. We refer to the guarantee in Claim F.2 for the precise functional form of this dependence, which quantifies how the problem becomes more difficult as $\alpha$ increases.

---

> > ### Comment · Reviewer_HXFc · 2025-04-03
> >
> > I have carefully read the reply from the authors. Thanks. My major concerns regarding the acceptance of the paper still remains:
> >
> > - lack of experimental contribution, I don't agree that simple experiments with robust statistics algorithms can't be done. see arXiv:2002.01432 or arXiv:1506.02428 just for example;
> > - the authors could not answer my question regarding the explicit dependence on alpha,
> > - I still feel that the technical contribution is quite limited. In particular, the 1D case has been well-understood which this paper falls back to as a black-box after a dimension reduction step. The data-dependent dimension reduction step itself goes back to earlier works such as [arXiv:2006.12476] and its following works.
> >
> > I will therefore maintain my assessment.

---

> > > ### Author Response · Authors · 2025-04-08
> > >
> > > We thank the reviewer for their time and for expressing their opinion on our submission. We respond to each point made by the reviewer (in some cases reiterating points made in our first response).
> > >
> > > **Experiments**: We note that the ICML call for papers welcomes papers in the “Theory of Machine Learning” category. Typically, such papers design algorithms with provable performance guarantees (as our work) without providing experimental evaluations of these algorithms. In other words, the lack of experiments per se should not be counted against a learning theory submission. That said, we believe that experimental evaluation of our algorithm is possible, and would be interesting to do in future work.
> > >
> > > **Dependence on $\alpha$**: Reiterating our first response, our main theorem was stated for $\alpha \leq 0.49$. Since the problem becomes harder as $\alpha$ increases, the worst-case occurs when $\alpha = 0.49$. Hence the “dependence on $\alpha$” is a universal constant that appears in the big-O of the sample complexity. If the reviewer was asking what happens when $\alpha$ becomes arbitrarily close to $1/2$, then we point out that the sample complexity would have a term that scales with $1/(1-2\alpha)$.
> > >
> > > **Technical Contribution**: The reviewer’s first concern appears to be based on the fact that the one-dimensional version of our problem was previously solved, and that we use a dimension-reduction procedure to reduce to that case. The reviewer is also concerned about similarities with the dimension-reduction method in [arXiv:2006.12476].
> > >
> > > On the first point, we would like to remark that essentially every single paper in algorithmic robust statistics over the past decade has developed efficient algorithms for high-dimensional robust estimation tasks whose one-dimensional versions were already solved. The whole contribution is the design of a computationally efficient algorithm in high dimensions. As an example, Lai et al. (2016)---one of the first papers that initiated the field—as well as many more recent works, also reduces to the one-dimensional case.
> > >
> > > On the second point: The similarity of our method to the method in [arXiv:2006.12476] (which studies a supervised learning problem without corruptions) is that we look at large eigenvalues of some kind of moment matrix in order to do dimension-reduction. This is in fact a classical generic idea in the literature that did not originate with [arXiv:2006.12476]. That said, that is where the similarities end. Specifically:
> > > * In our setting, we needed to develop a carefully tuned *exponential term* in our moment computations to dilute the effect of outliers without substantially altering the contributions from the non-outliers. This idea is new and it is crucial for the correctness of our algorithm. This exponential reweighting was developed specifically to leverage the structure of the outliers in our contamination model.
> > > * The aforementioned exponential reweighting meant that our “mean” was no longer an unbiased estimator of the true mean. To address this, we needed to develop a new analysis to show that the component of the bias in the direction of small eigenvalues was small.
> > > * In order to develop a sample-near optimal algorithm, we needed to do a much more careful analysis of the convergence of the error matrix.
> > > * Importantly, our result requires a *recursive* dimension-reduction rather than a one-shot method.
> > > * Once we have reduced to a low-dimensional problem, we need a vastly different method of solving the lower dimensional problem efficiently.

---

### Official Review · Reviewer_GjRk · 2025-03-09

**Overall Recommendation:** 4

**Summary:**

The authors consider the problem of robust mean estimation of an identity covariance Gaussian in the presence of mean-shift contamination. They specifically give the first computationally efficient algorithm for high-dimensional robust mean estimation with mean-shift contamination. Their algorithm has near-optimal sample complexity, runs in sample-polynomial time, and approximates the target mean to any desired accuracy. It does so by using a data dependent dimension reduction technique which recovers a low-dimensional subspace on which μ has a large projection, and then running a low-dimensional robust estimator on that subspace.

**Claims And Evidence:**

The claims made in the submission supported by clear and convincing evidence

**Essential References Not Discussed:**

According to my knowledge, there are no essential works not currently cited in the paper.

**Experimental Designs Or Analyses:**

There are no experimental designs in the paper.

**Methods And Evaluation Criteria:**

The proposed methods make sense for the problem or application at hand.

**Other Comments Or Suggestions:**

In the paper, it is implicitly assumed that || \mu || =O(1) while, n general, || \mu || can be O(d). The authors should elaborate on why that is a reasonable assumption, and what happens to the algorithm in case this assumption doesn't hold.

**Other Strengths And Weaknesses:**

The strength is the algorithm construction, which makes it run in sample polynomial time. The discussion on the algorithm construction is thorough, breaking it down to multiple intuitive steps.

**Questions For Authors:**

I have no questions for the authors.

**Relation To Broader Scientific Literature:**

The paper formulates the first computationally algorithm for high-dimensional robust mean estimation that runs in time polynomial in the sample complexity.

**Theoretical Claims:**

I checked the correctness of the claims in the main body and the full proof of Theorem 1.2 in the appendix, skipping all other technical claims in the appendix.

---

> ### Author Rebuttal · Authors · 2025-04-01
>
> We thank the reviewer for their time and for their positive assessment of our work. Below, we provide a clarification in response to their comment.
>
> The introductory proof sketch mentions that we can assume $\lVert\mu\rVert = O(1)$ due to naive outlier removal. This means that even if $\lVert\mu\rVert$ is arbitrary, we can preprocess the data to reduce it to the case where $\lVert\mu\rVert = O(1)$; a step that is formally carried out in the subsequent sections containing the full proofs. The argument is quite simple: At the start of the algorithm (line 6 of Algorithm 1), we perform a preprocessing step to obtain a rough estimate $\hat{\mu}_0$ satisfying $\lVert\hat{\mu}_0 - \mu\rVert = O(1)$. This can be achieved using any black-box outlier removal estimator designed for the Huber model from prior work. We then draw a new dataset and replace each point $x$ in that dataset by $x - \hat{\mu}_0$. This transformation effectively shifts the data distribution so that its true mean becomes $\mu - \hat{\mu}_0$, which has norm $O(1)$. After the algorithm produces an $\epsilon$-approximation $\hat{\mu}_1$ for this shifted mean, we simply add back $\hat{\mu}_0$ to $\hat{\mu}_1$, obtaining an $\epsilon$-approximation of $\mu$ (line 26 of Algorithm 1).

---

### Official Review · Reviewer_BKkQ · 2025-03-12

**Overall Recommendation:** 4

**Summary:**

This paper continues the recent line of work in robust mean estimation in high dimensions, with focus on getting arbitrary small (eps) error in the mean despite an alpha-fraction (for alpha in [0, 0.49]) of outliers.  The more commonly studied Huber model for outliers contamination can only achieve O(alpha) error, so this method has more requirements:
  - the inliers are of form N(mu, I) [this is very common], and outliers x_i of form N(z_i, I),  notably with same covariance as inliers
  - the n = Omega(2^O(1/eps^2)) samples are required

The fact the outliers have the same covariance is outliers seems like a mild assumption, and could maybe be relaxed more; this seems mostly to show that outliers are full rank and their covariance is psd.  But the form considered is natural.
The size of the samples is a large increase from Theta(d / eps^2) for robust methods that obtain O(alpha) error (with eps <= alpha).  But the 2^Omega(1/eps^2) appears necessary (from Kotekal+Gao 2025) for even d=1.

Moreover this second point allows that if we can reduce the problem to k=1/eps^2 dimensions, then we can net the space of directions (with 2^O(1/eps^2) directions), apply the 1-d version in each one, and return a point in the intersection of these intervals.  This is what the proposed method does.

What remains is to project to k dimensions so that the true inlier mean incurs at most eps error in this projection.  This is done iteratively somehow using the now-standard insight that the directions with most covariance contain both the true inlier mean, and any meaningfully biased set of outliers.  As such, the algorithm iteratively finds the top eigenvectors of the covariance matrix, and projects onto them -- reminiscent of Lai, Rao, Vempala 2016.
Although, some new ideas seem needed here -- especially ones that make use of the specific structure of the outlier distributions.

There are numerous details to get this to work, and tighten the runtime.  The details are technical, and appear to be handled correctly, but do not seem to use wholesale new insights.

**Claims And Evidence:**

Yes

**Essential References Not Discussed:**

No

**Experimental Designs Or Analyses:**

This is a theory paper, and not experiments are presented.  That is not a problem.

**Methods And Evaluation Criteria:**

Yes

**Other Comments Or Suggestions:**

N/A

**Other Strengths And Weaknesses:**

Strength:  I think the arbitrary smaller error variant of this problem is interesting to study.  This paper makes a major advancement in that sense going from d=1 to high dimensions d.

Weakness:  the runtime of 2^O(1/eps^2) is quite large, and makes the ultimate result of less general relevance.  Albeit, recent prior work claims this is needed even in d=1.

**Questions For Authors:**

N/A

**Relation To Broader Scientific Literature:**

The paper seems to have done a good job.

**Theoretical Claims:**

I followed the main arguments, and feel reasonable confident that they work.  But I did not verify all of the details.

---

> ### Author Rebuttal · Authors · 2025-04-01
>
> We thank the reviewer for their effort and their positive assessment of our work. We respond to the comments below:
>
> * (**Comparison with Lai, Rao, and Vempala 2016**) As we discuss in detail in Appendix (lines 662–671), although Lai, Rao, and Vempala (2016) iteratively reduce the dimension using the centered second moment of the data, this approach can only achieve an error of order $\alpha$, at best (in fact, their algorithm additionally incurs error with a logarithmic dependence in the dimension). Our dimension-reduction technique is fundamentally different, with the key innovation being the use of exponential reweighting for each sample. This mitigates the impact of outliers (in our model) and enables consistent estimation. Implementing this approach introduces additional challenges that must be addressed: (i) our dimension reduction method requires the space to have at least $1/\epsilon^2$ dimensions (as opposed to just one dimension in Lai, Rao, and Vempala 2016); (ii) the parameter used in exponential reweighting influences both the centering applied in the computation of the centered second moment and the sample complexity. As explained in Section 1.2, the first attempt for setting this parameter correctly does not immediately allow for reduction all the way down to the desired level. Instead, our algorithm operates in two stages: the first stage performs the initial dimension reduction from the last sentence, and the second stage resets that parameter to a refined value that allows for further reduction.
> * (**Runtime**) As the reviewer points out, the runtime of $2^{O(1/\epsilon^2)}$ is unavoidable, even in one dimension. This is because, even in the one-dimensional case, estimating the mean up to error $\epsilon$ information-theoretically requires $2^{\Omega(1/\epsilon^2)}$ samples, as shown in Kotekal & Gao (2025). The key conceptual point of our paper is that, prior to our work, it was unknown whether the high-dimensional problem incurs a significantly higher runtime (e.g., $2^{\Omega(d/\epsilon^2)}$). We rule out this possibility by showing that runtime $poly(d)2^{\Omega(1/\epsilon^2)}$ is enough (i.e., the dependence on the dimension is polynomial instead of exponential). Notably, this implies that setting $\epsilon = \Theta(1/\sqrt{\log d})$ allows for error tending to zero as $d \to \infty$) within an overall $\text{poly}(d)$ runtime. In contrast, every prior algorithm for the Huber model cannot achieve an error smaller than $\Omega(\alpha)$, which could be an absolute constant.

---

### Official Review · Reviewer_oS19 · 2025-03-14

**Overall Recommendation:** 4

**Summary:**

This work studies a relaxed model of robust mean estimation of a spherical Gaussian with identity covariance, where the adversarial points do not follow arbitrary distribution Q, but are sampled from $\mathcal{N}(z_i, I)$, where $z_i$ is an arbitrary point. Authors show that in such model consistent estimation is possible, and provide a computationally efficient algorithm for high-dimensional robust mean estimation.

**Claims And Evidence:**

Yes

**Essential References Not Discussed:**

No to the best of my knowledge.

**Experimental Designs Or Analyses:**

Not applicable

**Methods And Evaluation Criteria:**

Yes

**Other Comments Or Suggestions:**

I suggest adding more details on why the $2^{O(1 / \varepsilon^2)}$ term is necessary in the sample complexity. I do not think it is true in general (see Questions).

**Other Strengths And Weaknesses:**

Strengths: studies an interesting setting of robust mean estimation where consistent estimators exist. Proposes a computationally efficient algorithm for the case where inlier samples follow $\mathcal{N}(\mu, I)$.

Weaknesses:
1. The work only covers identity covariance case.
2. The dependancy on the corruption rate $\alpha$ does not appear in the main result.

**Questions For Authors:**

It seems that the work implicitly assumes that $\alpha = \Omega(1)$?
What happens when $\alpha = 0$ or $\alpha = \varepsilon$?
In both cases there exists much more sample efficient algorithms then what is provided by the authors.

**Relation To Broader Scientific Literature:**

This work extends the literature on robust mean estimation, studies a high-dimensional setting of the mean-shift contamination. This is the first result which looks at the computationally efficient algorithms for such contamination model.

**Theoretical Claims:**

Yes

---

> ### Author Rebuttal · Authors · 2025-04-01
>
> We thank the reviewer for their time, effort and their clarifying questions. We comment on the points mentioned and respond to the reviewers questions below:
>
> (**Identity covariance**) As pointed out by the reviewer, our algorithm works under the assumption that the covariance of the inliers is known. Please note that our algorithm does not require the covariance to be exactly equal to the identity; it is sufficient for the covariance to be *known*. (As pointed out in lines 85-88, for known covariance, by applying an appropriate transformation to the samples, we can reduce the problem to the identity covariance case.) We study the known covariance case, as it is the most fundamental setting for which we demonstrate the main conceptual point of the paper, which is an examination of how more structured adversaries in robust statistics can alter the statistical and computational landscape of estimation. As discussed in Section 3, extending our algorithmic result to the case of unknown covariance is an interesting problem for future work. There are several nuances in defining the problem for the unknown covariance case. For the sake of space, we point the reviewer to lines 629-654 for a thorough discussion.
>
> (**Dependence on alpha**). We discuss the two relevant points below:
> * (*Dependence in the main theorem*). The result in Theorem 1.2 holds for any $\alpha < 0.49$ (where $0.49$ can be replaced by any constant near $1/2$, as shown in Appendix F). The main conceptual point of the theorem is that even when nearly half of the samples are outliers, the algorithm efficiently achieves any *desired* error $\epsilon$ with the stated sample complexity of roughly $d/\epsilon^{2 + o(1)} + 2^{O(1/\epsilon^2)}$. This sample complexity is tight in this setting. Any prior algorithm would either require exponential runtime or incur a significantly larger error of $\Omega(1)$ (see lines 72-80 of the paper where this is explained), whereas our result allows for arbitrarily small error. Regarding other regimes for $\alpha$, prior work on one dimension (Kotekal & Gao (2025)) has fully specified the sample complexity as both a function of $\alpha$ and $\epsilon$. It would be interesting to investigate the exact dependence in high-dimensions. However, as with most prior work on robust statistics, the most interesting regime is to obtain a result for when the fraction of outliers $\alpha$ is a positive constant. This is because this is the hardest scenario for estimation (with the smallest amount of information about the target mean).
> * (*Special cases for $\alpha$*). Formally speaking, setting $\alpha = \epsilon$ is insufficient to obtain error $\epsilon$ using a Huber robust estimator, as any estimator for the Huber model incurs an error of at least $C\epsilon$ for some constant $C > 1$ (as established in the robust statistics literature). Ignoring this nuance regarding absolute constants, the regime $\alpha = \Theta(\epsilon)$ (and even more so $\alpha = 0$) corresponds to special cases where existing algorithms can be directly applied to achieve error $O(\epsilon)$—specifically, any algorithm designed for Huber contamination or the sample mean setting, respectively. Thus, these cases are orthogonal to our work as they are special cases that can be handled by existing algorithms. As mentioned earlier, the main question we focus on is obtaining a result for the case when $\epsilon \ll \alpha$, i.e., we aim to estimate with error significantly smaller than the fraction of outliers. This is an objective that was impossible to achieve with algorithms designed for the Huber model, and our algorithmic result for constant fraction of outliers has tight sample complexity.

---

### Decision · Program_Chairs · 2025-05-01

**Decision:**

Accept (poster)

**Comment:**

**Summary.**

The topic of this paper is robust mean estimation of a Gaussian distribution (multi-dimensional, with known covariance) when samples are contaminated in the sense where a (constant) proportion alpha comes from another Gaussian with (identical covariance matrix but) adversarially chosen mean (this is different from the more challenging Huber contamination model where the contamination distribution is arbitrary).

The main contribution is the design of an algorithm that:

* Can approximate the mean to arbitrary accuracy.
* Is agnostic to the contamination parameter alpha.
* Has computational complexity that is polynomial in the dimension and the number of samples.
* Has sample complexity that is not significantly higher than in one dimension.

**Strengths.**

* What is interesting about this setting is that contrary to the Huber contamination model, it is here possible to reach an arbitrary error $\varepsilon$ (the question is intriguing).
* In high dimensions, the authors propose the first computationally efficient algorithms for the proposed contamination model (algorithmic novelty).
* New ideas were needed in their analysis (BKkQ) based on the structure of the outliers’ distribution (sufficient novelty in the proofs).

**Weaknesses.**

* The sample complexity is high (unavoidably so, refer to Kotekal & Gao (2025)) which undermines the practicality of the method.
* The lack of numerical experiments was pointed out (HXFc); although I do not consider this to be a strict requirement for a strong theory paper.

**Discussion and reviewer consensus.**

After the rebuttal and discussion, there is consensus among the reviewer that the paper should be accepted.

**Additional remarks.**

HXFc originally had an incrementality concern. However, it was dispelled by the author response.

**Overall recommendation.**

This is serious work; the authors make progress on a very fundamental problem in statistics.
I don't think the lack of experiments should disqualify the paper. I recommend acceptance.